

# Parameterizations of US wildfire and prescribed fire emission ratios and emission factors based on FIREX-AQ aircraft measurements

Georgios I. Gkatzelis [1, 2, †], Matthew M. Coggon [2], Chelsea E. Stockwell [1, 2], Rebecca S. Hornbrook [3], Hannah Allen [4], Eric C. Apel [3], Katherine Ball [4], Megan M. Bela [1, 2], Donald R. Blake[5], Ilann Bourgeois [1, 2], Steven S. Brown [2, 6], Pedro Campuzano-Jost [1, 6], Jason M. St. Clair [7, 8], James H. Crawford [9], John D. Crounse [10], Douglas A. Day [1, 6], Joshua P. DiGangi [9], Glenn S. Diskin [9], Alan Fried [11], Jessica B. Gilman [2], Hongyu Guo [1, 6], Johnathan W. Hair [9], Hannah S. Halliday [9, ‡], Thomas F. Hanisco [7], Reem Hannun [7, 12], Alan Hills [3], L. Gregory Huey [13], Jose L. Jimenez [1, 6], Joseph M. Katich [1, 2], Aaron Lamplugh [1, 2], Young Ro Lee [13], Jin Liao [7, 14], Jakob Lindaas [15, §], Stuart A. McKeen [1, 2], Tomas Mikoviny [16], Benjamin A. Nault [1, 6, ‖], J. Andrew Neuman [1, 2], John B. Nowak [9], Demetrios Pagonis [1, 6, ¶], Jeff Peischl [1, 2], Anne E. Perring [1, 17], Felix Piel [16, 18, 19], Pamela S. Rickly [1, 2], Michael A. Robinson [1,2,6], Andrew W. Rollins [2], Thomas B. Ryerson [2, ^], Melinda K. Schueneman [1, 6], Rebecca H. Schwantes [2], Joshua P. Schwarz [2], Kanako Sekimoto [20], Vanessa Selimovic [21], Taylor Shingler [9], David J. Tanner [13], Laura Tomsche [9, 22, *, #], Krystal T. Vasquez [10], Patrick R. Veres [2], Rebecca Washenfelder [2], Petter Weibring [10], Paul O. Wennberg [10, 23], Armin Wisthaler [16, 18], Glenn M. Wolfe [7], Caroline C. Womack [1, 2], Lu Xu [10, +], Robert J. Yokelson [21], Carsten Warneke [2]

[1] Cooperative Institute for Research in Environmental Sciences, University of Colorado Boulder, Boulder, CO, USA
[2] NOAA Chemical Sciences Laboratory (CSL), Boulder, CO, USA
[3] Atmospheric Chemistry Observations & Modeling Laboratory, NCAR, Boulder, CO, USA
[4] Division of Chemistry and Chemical Engineering, California Institute of Technology, Pasadena, CA, USA
[5] Department of Chemistry, University of California, Irvine, CA, USA
[6] Department of Chemistry, University of Colorado Boulder, Boulder, CO, USA
[7] Atmospheric Chemistry and Dynamics Laboratory, NASA Goddard Space Flight Center, Greenbelt, MD, USA
[8] Joint Center for Earth Systems Technology, University of Maryland Baltimore County, Baltimore, MD, USA
[9] NASA Langley Research Center, Hampton, VA, USA
[10] Division of Geological and Planetary Sciences, California Institute of Technology, Pasadena, CA, USA
[11] Institute of Arctic & Alpine Research, University of Colorado, Boulder, CO, USA
[12] Joint Center for Earth Systems Technology, University of Maryland Baltimore County, Baltimore, MD, USA
[13] School of Earth and Atmospheric Sciences, Georgia Institute of Technology, Atlanta, GA, USA
[14] Universities Space Research Association, Columbia, MD, USA
[15] Colorado State University, Department of Atmospheric Science, Fort Collins, CO, USA
[16] Department of Chemistry, University of Oslo, Oslo, Norway
[17] Department of Chemistry, Colgate University, Hamilton, NY, USA
[18] Institut für Ionenphysik und Angewandte Physik, Universität Innsbruck, Innsbruck, Austria
[19] IONICON Analytik GmbH, Innsbruck, Austria
[20] Graduate School of Nanobioscience, Yokohama City University, 22-2 Seto, Kanazawa-ku, Yokohama, Kanagawa, Japan
[21] Department of Chemistry and Biochemistry, University of Montana, Missoula, MT, USA
[22] Universities Space Research Association, Columbia, MD, USA
[23] Division of Engineering and Applied Science, California Institute of Technology, Pasadena, CA, USA
Currently at:
† Institute of Energy and Climate Research, IEK-8: Troposphere, Forschungszentrum Jülich GmbH, Jülich, Germany
‡ U.S. Environmental Protection Agency, Research Triangle Park, NC, USA
§ AGI / AAAS Congressional Science Fellow
‖ Center for Aerosol and Cloud Chemistry, Aerodyne Research Inc., Billerica, MA, USA
¶ Weber State University, Ogden, UT, USA
^ Scientific Aviation, Boulder, CO, USA
* Institute of Atmospheric Physics, German Aerospace Center, Wessling, Germany



# Johannes Gutenberg University, Mainz, Germany
+ Cooperative Institute for Research in Environmental Sciences, University of Colorado Boulder, Boulder, CO,
USA and NOAA Chemical Sciences Laboratory (CSL), Boulder, CO, USA
*Correspondence to*: (g.gkatzelis@juelich.de and matthew.m.coggon@noaa.gov)
**Abstract.**
Extensive airborne measurements of non-methane organic gases (NMOGs), methane, nitrogen oxides, reduced
nitrogen-species, and aerosol emissions from US wild and prescribed fires were conducted during the 2019
NOAA/NASA Fire Influence on Regional to Global Environments and Air Quality campaign (FIREX-AQ). Here,
we report the atmospheric enhancement ratios (ERs) and inferred emission factors (EFs) for compounds measured
onboard the NASA DC-8 research aircraft for nine wildfires and one prescribed fire, which encompass a range of
vegetation types.
We use photochemical proxies to identify young smoke and reduce the effects of chemical degradation on our
emissions calculations. ERs and EFs calculated from FIREX-AQ observations agree within a factor of 2 with values
reported from previous laboratory and field studies for more than 80% of the carbon- and nitrogen-containing
species. Wildfire emissions are parameterized based on correlations of the sum of NMOGs with reactive nitrogen
oxides ($NO_y$) to modified combustion efficiency (MCE) as well as other chemical signatures indicative of
flaming/smoldering combustion, including carbon monoxide (CO), nitrogen dioxide ($NO_2$), and black carbon
aerosol. The sum of primary NMOG EFs correlates to MCE with an $R^2$ of 0.68 and a slope of -296 ± 51 g $kg^{-1}$,
consistent with previous studies. The sum of the NMOG mixing ratios correlates well with CO with an $R^2$ of 0.98
and a slope of 137 ± 4 ppbv of NMOGs per ppmv of CO, demonstrating that primary NMOG emissions can be
estimated from CO. Individual nitrogen-containing species correlate better with $NO_2$, $NO_y$, and black carbon than
with CO. More than half of the $NO_y$ in fresh plumes is $NO_2$ with an $R^2$ of 0.95 and a ratio of $NO_2$ to $NO_y$ of 0.55 ±
0.05 ppbv $ppbv^{-1}$, highlighting that fast photochemistry had already occurred in the sampled fire plumes. The ratio
of $NO_y$ to the sum of NMOGs follows trends observed in laboratory experiments and increases exponentially with
MCE, due to increased emission of key nitrogen species and reduced emission of NMOGs at higher MCE during
flaming combustion. These parameterizations will provide more accurate boundary conditions for modeling and
satellite studies of fire plume chemistry and evolution to predict the downwind formation of secondary pollutants,
including ozone and secondary organic aerosol.



## 1 Introduction

Open biomass burning in the form of wildfires, prescribed forest management fires, and agricultural burns is one of the largest sources of trace gases and aerosols worldwide (Akagi et al., 2011; Crutzen and Andreae, 1990). It is the dominant global source of black carbon and primary organic aerosol (Bond et al., 2013), and accounts for more than 20% of the global emissions of nitric oxide (NO) and carbon monoxide (CO) (Olivier et al., 2005; Yokelson et al., 2008; Wiedinmyer et al., 2011). It is the second largest global source of non-methane organic gases (NMOGs) (Akagi et al., 2011), and a major source of greenhouse gases, including methane ($CH_4$), carbon dioxide ($CO_2$), and nitrous oxide ($N_2O$) that impact the atmospheric carbon budget and climate (Sudo and Akimoto, 2007; Ward et al., 2012; Tian et al., 2016; Le Quéré et al., 2018).

During the last decade, the number of wildfires and prescribed fires in the US has sometimes exceeded 74,000 and 450,000 $yr^{-1}$, respectively (National Interagency Fire Center). Warming temperatures, drier climate, and a history of fire suppression are projected to increase the frequency and intensity of wildfires and lengthen fire seasons globally (Spracklen et al., 2009; Kloster et al., 2010; Pechony and Shindell, 2010; Moritz et al., 2012; Flannigan et al., 2013; Mann et al., 2016; Balch et al., 2017), which is already evident in the western US, Canada, the eastern Mediterranean, Siberia, and Australia (Westerling et al., 2006; Keywood et al., 2013; Yue et al., 2015). Wildfires in the US largely occur in the western conterminous states and Alaska, and typically account for 12 to 40 thousand $km^2$ of the annual total area burned (National Interagency Fire Center). In the southeastern US, prescribed fires and agricultural burns are a common land management tool used to improve ecosystem health or facilitate planting crops (Wiedinmyer and Hurteau, 2010; Cochrane et al., 2012). Since prescribed fires in the southeast currently account for about 25 thousand $km^2$ per year on average (National Interagency Fire Center), it is also important to characterize their emissions.

While wildfires and prescribed fires are favorable for many ecosystem functions, the atmospheric impacts of fire on climate, air quality, and health are a major concern. Particles directly emitted or formed via chemical processes have direct and indirect effects on climate by influencing the regional and global radiation balance and impacting cloud properties and precipitation (Braga et al., 2017; Cecchini et al., 2017; Hamilton et al., 2018; Thornhill et al., 2018; Kodros et al., 2020). Global mortality from outdoor pollution due to biomass burning smoke accounts for 600,000 premature deaths per year (Johnston et al., 2012), with particulate matter (PM) and $O_3$ posing the greatest risk factors (Akagi et al., 2014; Dennekamp et al., 2015; Brey and Fischer, 2015; Knorr et al., 2017; Apte et al., 2018). In smoke plumes, $O_3$ and secondary organic aerosols are photochemically produced from the interplay of $NO_x$, NMOGs, and meteorology (Tsimpidi et al., 2017; Hodshire et al., 2019). An essential first step to elucidate the factors contributing to PM and $O_3$ pollution downwind fires is to quantify primary gas- and particle-phase emissions.

Numerous studies have quantified emission factors (EFs; grams emitted per kg of dry fuel burned) for various fuel types and different fire characteristics using ground-based or airborne measurements in close proximity to wildland/prescribed fire plumes (e.g., Stockwell et al., 2016; Liu et al., 2017; Peng et al., 2020; Mouat et al., 2021; Lindaas et al., 2021; Permar et al., 2021) or controlled laboratory burns (e.g., Stockwell et al., 2014; Koss et al., 2018; Selimovic et al., 2018). Literature reviews to combine these results have been periodically conducted (Andreae and Merlet, 2001; Akagi et al., 2011; Andreae, 2019), with the most recent by Prichard et al. (2020). Nevertheless, uncertainties in the process-level understanding and model representation of fire emissions, plume rise, and chemistry still exist, which influence model performance in accurately capturing downwind $O_3$ and secondary organic aerosol formation (Müller et al., 2016; Reddington et al., 2016; Shrivastava et al., 2017). These uncertainties can result from an insufficient understanding of the chemistry and total emissions of $NO_x$ and NMOGs across fuel types, ecosystems, and fire combustion conditions (Warneke et al., 2011; Yokelson et al., 2013; Hatch et al., 2017).

In this study, we calculate western US wildfire emission factors for a broad range of gas- and particle-phase species measured aboard the NASA DC-8 during the 2019 Fire Influence on Regional to Global Environments and Air Quality (FIREX-AQ) campaign, which included the most comprehensive payload to date for airborne sampling of biomass burning emissions. We compare our results to the most recent laboratory and airborne field studies, including the fire sciences laboratory component of FIREX-AQ (hereafter referred to as FireLab) (Koss et al., 2018), the fourth Fire Lab at Missoula Experiment, FLAME-4 (Stockwell et al., 2015), the Western Wildfire Experiment for Cloud Chemistry, Aerosol Absorption, and Nitrogen, WE-CAN (Permar et al., 2021), and the Studies of



Emissions and Atmospheric Composition, Clouds and Climate Coupling by Regional Surveys, SEAC[4]RS (Liu et
al., 2017; Wolfe et al., 2022), as well as results summarised in the review by Andreae (2019). We parameterize
wildfire emissions based on correlations of carbon- and nitrogen-containing species to CO, $NO_2$, black carbon, and
modified combustion efficiency (MCE) to improve future modeling efforts to accurately capture the chemical
evolution of wildfire smoke.
**2 Methods**
**2.1 Platforms and Instrumentation**
The NASA DC-8 aircraft was deployed with an extensive suite of instruments to measure the gas- and particle-
phase pollutants emitted and photochemically produced downwind of US wildfires. Figure 1 and Table 1 show the
research flights analyzed here to capture freshly emitted wildfire smoke from 22 July to 3 September 2019. In total,
16 crosswind plume transects downwind from 9 western wildfires and 1 eastern prescribed fire are analyzed, which
represent a range of fuel types, including timber, grass, dead trees, logging debris, brush, and litter. The transects
are selected based on aging proxies to examine emissions with minimal atmospheric processing. The physical age
is determined based on transect proximity to the fire, an estimated plume rise time, and wind speed (Holmes et al.,
2020) and ranged from 10–153 min (1–40 km) downwind for the plumes described here. The MCE, defined as
$\Delta CO_2/(\Delta CO_2+\Delta CO)$, is commonly reported to quantify the fire conditions and describes the relative amount of
flaming and smoldering combustion (Yokelson et al., 1996). Pure flaming fires have an MCE near 0.99, while
smoldering fires vary over a wider range but are most often near 0.8 (Akagi et al., 2011). For the freshest plume
crossings, the MCE was on average 0.90 ± 0.04 (range 0.94–0.85), suggesting a mix of flaming and smoldering
emissions.

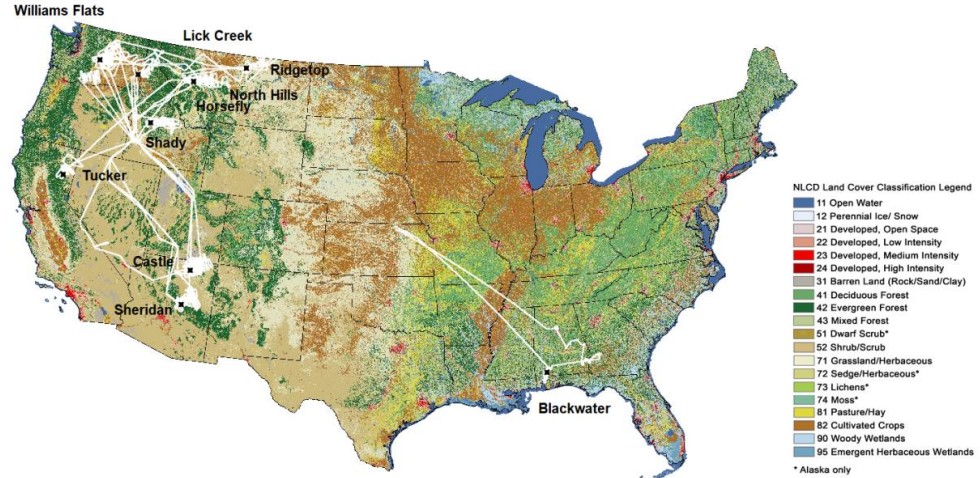


**Figure 1:** Selected NASA DC-8 flight tracks for sampling the wildfire and prescribed fire plumes during the 2019 FIREX-AQ.
Fires discussed in this study are denoted by black markers. The US map is colored by land cover classification according to the
2019 National Land Cover Database (https://www.mrlc.gov/).
Multiple instruments performed fast response *in situ* measurements of gas- and particle-phase species summarized
in Table 2. The University of Colorado aircraft aerosol mass spectrometer (CU HRAMS, AMS in the following)
(Canagaratna et al., 2007; Guo et al., 2021) measured organic aerosol, particulate ammonium, and nitrate ($pNO_y$)
that consisted of inorganic nitrates ($pNO_3$), organic nitrates ($pRONO_2$), and nitroaromatics ($pArNO_2$) (Day et al.,
2022). Black carbon aerosol concentration was measured by a Single-Particle Soot Photometer (SP2) and scaled
(~10%) to represent the total accumulation-mode (Schwarz et al., 2008). NMOGs were measured by the NOAA
proton transfer reaction time-of-flight mass spectrometer (PTR-ToF-MS) (Yuan et al., 2016), two whole-air



samplers, namely the NOAA integrated Whole Air Sampler (NOAA iWAS) (Lerner et al., 2017) and the University
of California, Irvine Whole Air Sampler (UCI WAS) (Colman et al., 2001; Simpson et al., 2020), the NCAR Trace
**Table 1:** Freshest plume crossings identified for analysis during FIREX-AQ 2019.

| Fire | Transect number | Date and Time, UTC | Fuel Type | Maleic anhydride to furan as an indicator of OH exposure (ppb ppb⁻¹) | Physical age (s) | MCE |
|---|---|---|---|---|---|---|
| **Shady** | 0 | 7/25/2019 22:48 | Understory: Ponderosa pine, white-Douglas fir, quaking aspen, two-needle pinyon-Utah juniper forest w/ open shrubs, grasses, and timber litter | 0.09 | 1350 | 0.91 |
| **Shady** | 9 | 7/25/2019 23:47 | | 0.07 | 1250 | 0.90 |
| **North Hills** | 0 | 7/29/2019 23:21 | Savanna: Ponderosa pine savanna, Douglas-fir-Pacific ponderosa pine, ocean spray forest with Idaho fescue-bluebunch wheatgrass | 0.13 | 600 | 0.86 |
| **Tucker** | 0 | 7/30/2019 2:40 | Shrubland: Sagebrush-greasewood shrubland with open grasses | 0.12 | 1720 | 0.91 |
| **Ridgetop** | 4 | 8/2/2019 23:18 | Grassland: Bluebunch wheatgrass, bluegrass with sagebrush-greasewood shrubs and savanna | 0.14 | 2620 | 0.94 |
| **Lick Creek** | 1 | 8/3/2019 1:13 | Forest: Grand-Douglas fir, Pacific ponderosa pine, ocean spray forest | 0.15 | 1500 | 0.91 |
| **Williams Flats** | 0 | 8/3/2019 22:22 | Grassland: Idaho fescue-bluebunch wheatgrass-cheatgrass, sagebrush shrublands under open Douglas-fir-Pacific ponderosa pine, ocean spray savanna/forest | 0.16 | 890 | 0.91 |
| **Williams Flats** | 21 | 8/4/2019 0:41 | | 0.14 | 6130 | 0.91 |
| **Horsefly** | 1 | 8/6/2019 23:20 | Forest: Managed: Subalpine-Douglas fir, lodgepole-whitebark-Pacific ponderosa-Mature lodgepole pine, Engelmann spruce oceanspray forest | 0.12 | 3890 | 0.87 |
| **Horsefly** | 3 | 8/6/2019 23:28 | | 0.11 | 6250 | 0.85 |
| **Williams Flats** | 7 | 8/9/2019 1:49 | Forest: Douglas-fir-Pacific ponderosa pine, ocean spray forest with grassland understory | 0.11 | 5460 | 0.91 |
| **Castle** | 0 | 8/13/2019 0:18 | Forest: Ponderosa pine, two-needle-pinyon-Utah juniper, Douglas-white fir, Madrean pine-oak, quaking aspen forest | 0.15 | 1540 | 0.90 |
| **Castle** | 0 | 8/13/2019 23:17 | | 0.16 | 9200 | 0.90 |
| **Castle** | 10 | 8/14/2019 1:32 | | 0.07 | 1600 | 0.88 |
| **Sheridan** | 1 | 8/17/2019 0:42 | Forest: Pinyon-Utah juniper forest with Turbinella oak-alderleaf mountain mahogany shrubland | 0.15 | 1200 | 0.91 |
| **Blackwater** | 8 | 8/30/2019 17:11 | Forest: Prescription, primarily shrubs, grasses and litter from loblolly-longleaf-slash pine, willow-laurel-turkey-water oak, and magnolia forest | 0.31 | 580 | 0.93 |

**Table 2:** Descriptions of the instrumentation aboard the NASA DC-8 used in this study.

| Species Measured | Technique | Frequency [Hz] | Inlet Setup | Reference |
|---|---|---|---|---|
| O₃, NO, NO₂, NOᵧ | Chemiluminescence | 1 | PFA, approx. 1 m long, 1 slpm for each species; NO and NO₂ additionally pass through 50.9 cm³ quartz cells | *Ryerson et al.* (2000) |



| Species | Instrument | Frequency | Inlet | Reference |
|---|---|---|---|---|
| $CO_2$, CO, $CH_4$, $H_2O$ | 2x Laser Absorption Spectroscopy | 1-5 | ¼ in stainless steel, 2 m long, 3slpm flow | *Sachse et al.* (1991) *Bourgeois et al.* (2022) |
| $NH_3$, speciated hydrocarbons and OVOCs | PTR-ToF-MS | 1 ($NH_3$) 10 (others) | PFA, 2 m long, ~20 LPM (before Aug 3), ~60 LPM (from Aug 3 onwards), heated to 60°C | *Müller et al. (2016)* *(with modifications)* |
| PAN, PPN, other PANs | Chemical Ionization Mass Spectrometry (CIMS) | 1-10 | ½" FEP tubing | *Zheng et al.* (2011) |
| HONO, HCN, HNCO, HCOOH, $N_2O_5$, HPMTF, halogenated compounds | Iodide ToF-CIMS | 1 | PTFE, 1m long, 6 SLPM, heated to 40°C | *Veres et al.* (2020) |
| NO | Laser Induced Fluorescence | 1 | PFA and silcosteel, 1m length, unheated, overflow at 10-20 slm | *Rollins et al.* (2020) |
| $CH_2O$, $C_2H_6$ | Laser Absorption Spectroscopy | 1 | Heated HIAPER Inlet followed by several meters of heated PTFE Teflon tubing | *Richter et al.* (2015); *Fried et al.* (2020) |
| $C_2$-$C_{10}$ Alkanes, $C_2$-$C_4$ Alkenes, $C_6$-$C_9$ Aromatics, $C_1$-$C_5$ Alkylnitrates, etc. | Whole Air Sampling | Up to 168 per flight | stainless steel | *Simpson et al.* (2001) |
| Speciated hydrocarbons and OVOCs | $H_3O^+$ ToF-CIMS | 1-5 | PTFE, 1m long, 1-2 LPM, heated to 50°C | *Yuan et al.* (2016) |
| $C_2$-$C_{10}$ Alkanes, $C_2$-$C_4$ Alkenes, $C_6$-$C_9$ Aromatics, $C_1$-$C_5$ Alkyl nitrates, etc. | Whole Air Sampling | Up to 72 per flight | PFA, 2m Long, ~60 LPM, unheated | *Lerner et al.* (2017) |
| $C_3$-$C_{10}$ hydrocarbons, $C_1$-$C_7$ OVOCs, HCN, $CH_3CN$, halogenated VOCs, etc. | HR-ToF-GC/MS | 0.0095 | Restek Silcosteel, 2.5 LPM, heated to 40°C | *Apel et al.* (2010) |
| $CH_2O$ | Laser Induced Fluorescence | 1-10 | PFA and silcosteel, 1m length, unheated, overflow at 10-20 slm | *Cazorla et al.* (2015) |
| $H_2O_2$, organic peroxides, organic acids, isoprene oxidation products, etc. | CIMS | 1 | A glass tube (3 cm ID and 47cm long) coated with a thin layer of (Fluoropel PFC 801A, Cytonix Corp.). The tube is gently heated and the sampling flow rate through the glass tube is >=40 m/s. | *Crounse et al.* (2006) |
| glyoxal, methylglyoxal, HONO, $NO_2$ | Airborne Cavity Enhanced Spectrometer | 1 | PTFE teflon, <1 m length, inlet heated to 25°C, 10.5 vlpm | *Min et al.* (2016) |
| BC mass concentration | SP2 | 1 | NASA Langley inlet with optional dilution | *Schwarz et al.* (2008) |
| Submicron aerosol composition | CU-HR-AMS | 1 (up to 10 Hz in plumes) | HIMIL tall inlet, 1.3 m SS 0.18" ID+ 0.45 m 0.08" ID tubing + pressure controlled instrument inlet ( <0.3 s total residence time) | *Guo et al (2021); Canagaratna et al (2007)* |

Organic Gas Analyzer (Apel et al., 2015), a fast online gas chromatograph outfitted with a Time-of-Flight mass
spectrometer (TOGA-TOF), the Caltech chemical ionization time-of-flight mass spectrometer (CIT-ToF-CIMS),
and for selected flights the University of Innsbruck / University of Oslo (UIBK/UiO) PTR-ToF-MS (prototype PTR-
TOF 4000X2; IONICON Analytik GmbH, Innsbruck, Austria). Three instruments were used in this study that
measured formaldehyde: the In Situ Airborne Formaldehyde (ISAF) instrument (Liao et al., 2021), the Compact
Atmospheric Multispecies Spectrometer (CAMS) (Weibring et al., 2007), and the UIBK/UiO PTR-ToF-MS. ISAF
and CAMS correlated with an $R^2$ coefficient of 0.99 and a slope of 1.27, as discussed by Liao et al. (2021); whereas





the UIBK/UiO PTR-ToF-MS agreed better with the CAMS, with a slope of 1.02. In this study, we use the ISAF
measurements, which have the best time response compared to all other instruments and adjust the mixing ratios to
match those reported by CAMS and the UIBK/UiO PTR-ToF-MS. The NOAA Iodide ion chemical ionization mass
spectrometer (NOAA CIMS) (Veres et al., 2020) was used to measure formic acid (HCOOH), nitrous acid (HONO),
and dinitrogen pentoxide ($N_2O_5$). CO and $CH_4$ were measured via mid-IR wavelength modulation spectroscopy by
the Differential Absorption Carbon Monoxide Measurement (DACOM) instrument (Sachse et al., 1991). $CO_2$ was
measured via nondispersive infrared absorption spectroscopy using a LICOR model 7000 analyzer (Vay et al, 2009).
NO, $NO_2$, and $NO_y$ were measured by the NOAA chemiluminescence instrument (Bourgeois et al., 2020). $NO_y$
measures the sum of reactive nitrogen compounds, including NO, $NO_2$, HONO, peroxy nitrates, alkyl and
multifunctional nitrates, and particulate nitrate. Additional measurements of HONO and $NO_2$ were provided by the
NOAA Airborne Cavity Enhanced Spectrometer (ACES) (Min et al., 2016) and NO by the NOAA Laser Induced
Fluorescence instrument (NO-LIF) (Rollins et al., 2020). Glyoxal and methylglyoxal were measured by ACES, and
ammonia ($NH_3$) by the UIBK/UiO PTR-ToF-MS (Müller et al., 2016; Tomsche et al., 2023). The Georgia Tech
CIMS (GT-CIMS) was used to measure peroxyacetyl nitrate (PAN) and other PAN-like compounds such as
peroxylpropionyl nitrate, peroxyacryloyl nitrate, and peroxylbutyryl nitrate. Finally, the plume structure was
obtained from aerosol backscatter measured with the NASA Langley Airborne Differential Absorption Lidar
(DIAL). All measurements reported here are provided in the NASA FIREX-AQ data repository (NASA airborne
science data for atmospheric composition, 2019).
In this study, we focus on quantifying total and speciated NMOG emissions, which were predominantly measured
by PTR-ToF-MS, the two Whole Air Samplers, and TOGA-TOF. The same NOAA PTR-ToF-MS and the iWAS
systems were used at the US Forest Service's Missoula Fire Sciences Laboratory (FireLab) in 2016 as a precursor
to FIREX-AQ and described by Koss et al. (2018). Koss et al. (2018) speciated isomers measured by PTR-ToF-MS
using gas chromatography pre-separation and reported isomer distributions for over 150 individual masses. Here,
we compare these isomer distributions to the speciation derived based on the comparison of the GC-MS and PTR-
ToF-MS measurements conducted aboard the NASA DC-8 (Table S5). Two calibration methods were used to
determine NMOG sensitivities for the PTR-ToF-MS. For commercially available compounds, sensitivities were
determined by gravimetrically prepared standards or by liquid calibration, as described by Coggon et al. (2019).
Sensitivities for other species were estimated based on calculated proton transfer rate coefficients, as described by
Sekimoto et al. (2017). For the WAS system(s), NMOGs were calibrated using gravimetrically prepared standards,
as described by Lerner et al. (2017). A detailed description of the PTR-ToF-MS and WAS setups as well as NMOG
uncertainty is included in the supplement.
**3 Results and discussion**
**3.1 Plumes with minimal photochemical aging**
Emissions from wildfire plumes chemically transform once injected into the atmosphere (e.g., Akagi 2012;
Robinson et al., 2021; Decker et al., 2021; Xu et al., 2021). However, safety and operational constraints limit the
proximity of airborne sampling to the fire. An essential first step to quantifying wildfire primary emissions is to
identify plume samples that have undergone minimal chemical processing. Commonly, the freshest plumes are
identified using the plume age calculated from the distance downwind of the wildfire using the onboard measured
average wind speed (e.g., Permar et al., 2021) but neglecting plume rise. The physical age does not necessarily
identify plume crossings with the least chemical processing since the sampled smoke can be impacted by
meteorology, solar radiation, radical concentrations, and sampling artifacts related to the aircraft's position relative
to the center of the plume (Robinson et al., 2021; Decker et al., 2021; Wang et al., 2021).
Here, we account for oxidation by hydroxyl radical (OH) using the ratio of primary and secondary NMOG wildfire
tracers, specifically furan (a primary species; Koss et al., 2018) and maleic anhydride (a slow-reacting, secondary
species observed downwind of fires) (Zhao and Wang, 2017). Coggon et al. (2019) show that maleic anhydride
quickly forms downwind of fires from the OH oxidation of furans, and Wang et al. (2021) show that the distribution
of maleic anhydride in plumes closely mirrors the distribution of OH exposure. Since furan is a direct wildfire
emission and maleic anhydride is a chemical product of furan chemistry that is not significantly emitted from fires
(Coggon et al., 2019; Wang et al., 2021), the ratio of maleic anhydride to furan (MA/F) is expected to increase



downwind of a fire and exhibit a minimum in the least-processed plumes. This ratio is used as a photochemical clock to identify the freshest sampled plumes by extracting the lowest MA/F transect per wildfire plume and reduce the effects of chemical degradation on our primary NMOG emission calculations. We note that this technique may not account for the faster photolysis of light-absorbing species (such as HONO) or fast interconversion between NO and $NO_2$, but that the sum of reactive nitrogen species ($NO_y$) is expected to be conserved downwind of fires (Lindaas et al., 2020). Also, a quantitative relationship between MA/F and OH exposure is not presented here as the yield of maleic anhydride from furan oxidation requires further laboratory quantification.

Figure 2a shows the maleic anhydride, furan, and CO concentration downwind of the Williams Flats wildfire on August 3, 2019, as a characteristic example. Also shown are the MA/F and the median physical smoke age calculated for each plume crossing. Here we use the high time resolution of PTR-ToF-MS for MA and furan concentrations, but furan is additionally scaled by 0.46 to match the TOGA GC-MS concentrations as discussed in Sect. 3.2. VOC and CO concentrations were highest closer to the wildfire and decreased downwind, primarily due to dilution. The MA/F increased from 0.20 to 0.86 downwind of the fire, indicating active chemical conversion of furan to maleic anhydride. The physical smoke age followed the same increase from 0.5 to 4 hours. Figure 2b shows the MA/F for all wildfire plumes and the extracted freshest plume crossings. The freshest plume crossings had a median MA/F of 0.13 (0.1–0.16, 25th–75th), and their corresponding physical age was less than 1.46 h (0.6–1.74) (see Table 1). However, certain fires with similar MA/F ratios drastically ranged in physical age from 15 minutes to as high as 3–4 hours.

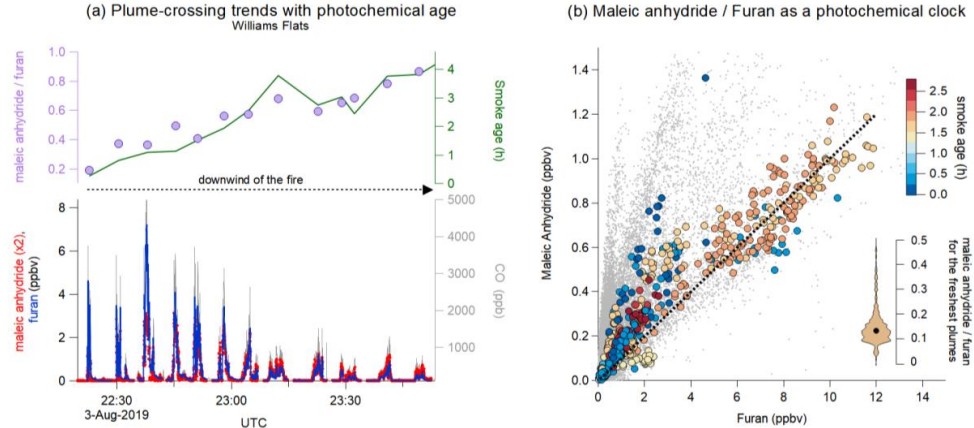

**Figure 2:** (a) Mixing ratios of maleic anhydride, furan, and CO (bottom) and ratios of maleic anhydride to furan (top) in 12 crosswind plume transects of smoke from the Willams Flats fire on 3 August 2019. The maleic anhydride to furan ratio increases as the plume ages during transport away from the Williams Flats. (b) Comparison of the maleic anhydride and furan mixing ratios used as a photochemical clock to identify the freshest plume crossings during FIREX-AQ. Grey points are all 1-sec resolution measurements during FIREX-AQ, and circles are the chosen freshest plume crossings colored by the physical smoke age. The violin plot shows the variability of the ratio of maleic anhydride to furan for the freshest wildfire transects.

Figure S1 further highlights differences in the physical and chemical age of a fire by focusing on the Williams Flats wildfire and the Blackwater prescribed fire. The DIAL image shows the shape and evolution of the wildfire smoke from overpass flights. For the Williams Flats fire, the DC-8 sampled emissions by performing raster patterns perpendicular to the smoke, whereas for the Blackwater fire, the DC-8 also flew along the smoke plume at various altitudes. For the Blackwater fire, the MA/F increased rapidly up to 1.4 ppbv ppbv$^{-1}$ 30 km downwind of the wildfire, while for the Williams Flats fire, the ratio reached a maximum of 1 ppbv ppbv$^{-1}$ 120 km downwind of the fire. These differences highlight the importance of accounting for the chemical rather than the physical age of a fire to determine the freshest transects. The MA/F for fresh, unaged smoke during the FireLab study was ~ 0.04 ppbv ppbv$^{-1}$ (Wang et al., 2021), showing that even the freshest plume transects sampled during FIREX-AQ were partially photochemically processed, with a median MA/F that was 0.14 ppbv ppbv$^{-1}$. Fire plumes sampled closest to the emission source that showed significant chemical processing with a MA/F > 0.20 are excluded from this analysis.



The exception is the Blackwater prescribed fire that was the only fire representative of southeastern US fuel types
included in our analysis, even though the freshest plume crossing had a MA/F of 0.3. Further evaluation of biases
during FIREX-AQ for fast-reacting species is discussed in Sect. 3.3.

**3.2 Instrument comparisons**

NMOG measurements obtained from the NOAA PTR-ToF-MS were compared to other instruments onboard the
DC-8, including TOGA-TOF, 2 WAS systems, CIT-CIMS, UIBK/UiO PTR-ToF-MS, and NOAA CIMS. Table S5
provides correlations of the PTR-ToF-MS measurements to other instruments. For calibrated compounds, the
NOAA PTR-ToF-MS and the UIBK/UiO PTR-ToF-MS agreed within 10–35% for methanol, acetonitrile, acetone,
methyl ethyl ketone (MEK), benzene, toluene, $C_8$ and $C_9$ aromatics, and monoterpenes. The NOAA PTR-ToF-MS,
and NOAA CIMS agreed within uncertainty for hydrogen cyanide (HCN), isocyanic acid (HNCO), and formic acid,
respectively. CIT-CIMS agreed with the NOAA PTR-ToF-MS for HCN whereas for phenol it was lower by a factor
2. Both instruments were calibrated for phenol suggesting that differences could be due to PTR-ToF-MS
fragmentation of higher molecular weight gases that produce signals at the phenol ion mass, or differences in the
detection of other isomers from the two instruments.
Although the PTR-ToF-MS provides high time resolution measurements, it cannot speciate NMOG isomers detected
at the same exact mass. In the following, we compare mixing ratios derived for the PTR-ToF-MS chemical formula
to the combined isomer signals derived from GC-MS, given in parentheses. When compared to the iWAS, WAS,
and TOGA-TOF measurements, the NOAA PTR-ToF-MS was within ±25–35% for $CH_4O$ (methanol), $C_2H_3N$
(acetonitrile), $C_2H_4O$ (acetaldehyde), $C_2H_6O$ (ethanol), $C_6H_6$ (benzene), $C_7H_8$ (toluene), $C_3H_3N$ (acrylonitrile),
$C_3H_4O$ (acrolein), $C_3H_6O$ (acetone + propanal), $C_8H_{10}$ (ethylbenzene + m-, p-, and o-xylenes), and $C_4H_6O$ (methyl
vinyl ketone + methacrolein + 2-butenal). However, the NOAA PTR-ToF-MS was higher by a factor of 2 or more
for $C_2H_6S$ (dimethyl sulfide), $C_4H_5N$ (pyrrole + butene nitrile isomers), $C_4H_4O$ (furan), $C_3H_6O_2$ (methyl acetate +
ethyl formate + hydroxyacetone), $C_5H_6O$ (2-methyl furan + 3-methyl furan), $C_5H_4O_2$ (furfural + 3-furaldehyde), and
$C_{10}H_{16}$ (monoterpenes) whereas $CH_3NO_2$ (nitromethane) agreed with the WAS but was lower than TOGA.
The discrepancies between the GC-MS techniques and PTR-ToF-MS for a number of key species, such as furans,
generally show that the PTR-ToF-MS measures more signal than what can be accounted for by GC-MS. This
observation likely results from a combination of (a) PTR-ToF-MS fragmentation of higher molecular weight gases
that produce signals at parent ion masses, (b) the detection of isomers that cannot elute through a GC column, and
(c) the detection of molecules that are lost to canister sampling. To investigate the causes of these discrepancies,
Table S5 shows isomer distributions for masses detected by the PTR-ToF-MS that are known to represent the sum
of two or more overlapping isomers. These isomer distributions are calculated from the ratio of GC-MS
measurements to the corresponding PTR-ToF-MS mass. Each ratio represents the fraction of the total signal
measured by PTR-ToF-MS that is associated with a given isomer. For example, GC-MS measurements identify 2-
methylfuran and 3-methylfuran as the key isomers with the molecular formula $C_5H_6O$. The slope of isomers to PTR-
ToF-MS measurements of $C_5H_6O$ represents the isomer fraction detected by PTR-ToF-MS.
The isomer distributions shown in Table S5 are compared to those reported for laboratory smoke by Koss et al.
(2018). Koss et al. (2018) assigned PTR-ToF-MS masses based on literature searches, intercomparisons of PTR-
ToF-MS measurements to other in situ instrumentation, and offline analysis by coupling GC effluent of sampled
smoke to the inlet of the PTR-ToF-MS (combined instrumental setup termed GC-PTR-ToF-MS). For low molecular
weight gases known to elute through a GC column, Koss et al. (2018) assigned isomer distributions based on the
total signal detected by GC-PTR-ToF-MS, which includes signals from parent ions produced from proton-transfer
as well as fragments from higher molecular weight gases that elute through a GC. For example, at $C_5H_6O$-H+ (m/z
83.0491), 51% of the signal resulted from the elution of 2-methylfuran, 9% resulted from 3-methylfuran, and 37%
was associated with other peaks in the chromatogram that produced signals at $C_5H_6O$-H+ (unidentified isomers +
fragments of higher masses). We note that the PTR-ToF-MS instrument employed in this study is the same as that
used by Koss et al. (2018) and is operated with the same drift field (E/N = 120 Td).
For species measured during FIREX-AQ where the PTR-ToF-MS reported significantly more mass than the GC
instruments, we find that the isomer distributions derived in this study significantly differ from those derived by





Koss et al (2018) (Table S5). This is most pronounced for the monoterpenes but also the furanoic species, such as
furan ($C_4H_4O$), methylfurans ($C_5H_6O$), and furfurals ($C_5H_4O_2$). Hatch et al. (2017) showed that more than 30
different isomers can contribute to the monoterpenes signal based on two dimensional GC. However, the
conventional GC instruments used during FIREX-AQ could only detect a fraction of these isomers. Furthermore,
differences in sensitivity for the different isomers would further increase the quantification uncertainties for both
GC and PTR-ToF-MS. For the furanoic masses, the PTR-ToF-MS measures a higher fraction of unknown isomers
and fragments than what is reported by Koss et al. (2018). This result holds whether comparing against isomer
distributions derived using TOGA (an online GC method) or WAS methods (a canister sampling method),
suggesting that uncertainties due to differences in calibration or canister effects are small. These results suggest that
the total signal of furans measured by PTR-ToF-MS during FIREX-AQ is likely influenced by gases that cannot
pass through a GC column, which includes the possibility of unidentified isomers and fragments from higher
molecular weight species. We note that this result is not specific to the PTR-ToF-MS used in this study, as the
agreement between the NOAA PTR-ToF-MS and UIBK/UiO PTR-ToF-MS for these masses is within 3% (Table
S5).
Furans are an important contributor to VOC reactivity and significantly contribute to the formation of ozone and
other secondary gases (Gilman et al. 2015, Hatch et al. 2017, Coggon et al. 2018). For models employing emission
factors of furans, we recommend using emission factors derived using GC-based methods given that multiple
isomers can be detected with PTR-TOF-MS at the furan mass. This also applies to other specific compound classes.
In Table S1, we include the methods used in this study to derive emission factors. For applications where the fast
time-resolution from PTR-ToF-MS is needed (e.g., in deriving cross-plume trends in gases), (Decker et al. 2021; Xu
et al. 2021), the interpretation of trends in furans should include the possibility of unknown isomers and fragments.

### 3.3 Emission ratios and emission factors of US wildfire smoke

The freshest plume transects are used to estimate the primary emissions for individual fires. Table 3 shows the
average compound-specific enhancement ratios to CO which we interpret as emission ratios (ERs) for most species,
and the inferred emission factors (EFs) calculated for more than 100 species and groups of species from the freshest
wildfire plume transects sampled during FIREX-AQ. ERs and EFs for each fire are also calculated and provided in
Tables S2 and S3. Given that fast chemistry already occurred in some fire transects, the ER and EF estimates of
highly reactive species like HONO are lower bounds. ERs are the slope of a linear fit of each species with CO
mixing ratios (see section S1). EFs were calculated following Eq. (1):
$$EF_i = F_C \cdot \frac{MM_i}{AW_C} \cdot \frac{\Delta i / \Delta CO}{\sum_{x=1}^{n} \left( NC_x \cdot \frac{\Delta C_x}{\Delta CO} \right)}, \qquad (1)$$
where $EF_i$ is the emission factor of compound $i$ calculated similarly to Akagi et al. (2011); $F_C$ is the carbon fraction
of the fuel assumed to be 0.5 g $g^{-1}$; $MM_i$ is the molar mass of $i$; $AW_C$ is the atomic mass of carbon (12 g $mol^{-1}$);
$\Delta i / \Delta CO$ is the emission ratio of a compound relative to CO; $NC_x$ is the number of carbon atoms in C-containing
species x, and $\Delta C_x / \Delta CO$ is the emission ratio of species x to CO. This method assumes that all the carbon lost from
the fuel as it burns is emitted and measured, which is a reasonable approximation as CO, $CO_2$, and $CH_4$ account for
most of the emitted carbon (Akagi et al., 2011). The denominator of the last term estimates total carbon relative to
CO. Species $C_x$ includes all species shown in Table 3. The carbon not quantified by the suite of instrumentation
available during FIREX-AQ likely results in emission factor overestimates no more than 1–2% (Yokelson et al.,
2013; Stockwell et al., 2015).
Figure 3 shows the average chemical composition of freshly emitted wildfire smoke in g $kg^{-1}$ (see Eq. (1)). $CO_2$,
CO, and $CH_4$ are 97% of the total mass. The remaining 3% consisted of gas- and particle-phase carbon-containing
(C-containing, 2.6%) and nitrogen-containing (N-containing, 0.3%) species. 50.4% and 0.7% of this remaining C-
containing total mass results from organic aerosol and black carbon (BC), respectively. In the gas phase, 6.4% of
the remaining C-containing species mass, which includes all species in Figure 3a, were phenolic compounds and
furans, 4% formaldehyde (HCHO), 4% glycolaldehyde and acetic acid ($C_2H_4O_2$), 3.7% acetaldehyde ($CH_3CHO$),
2.1% methanol, 5.8% remaining compounds with one oxygen atom ($C_xH_yO$), 6.9% remaining compounds with two
oxygen atoms ($C_xH_yO_2$), 3.1% aromatics, 6.3% alkenes, 2.8% alkanes, and 3.3% other species. N-containing species



mass, shown in Figure 3b, consisted of organic and inorganic nitrate, and other organic nitro compounds such as
nitroaromatics ($pNO_y$, 19%) and ammonium ($pNH_4^+$, 8.5%) in the particle-phase; whereas, the dominant gas-phase
N-containing species mass was from ammonia ($NH_3$, 18.5%), followed by nitrogen dioxide ($NO_2$, 17.5%), isocyanic
acid (HNCO, 8.5%), hydrogen cyanide (HCN, 5%), peroxyacyl nitrates (PANs, 7%), nitrous acid (HONO, 4.8%),
nitric oxide (NO, 2.5%), and others at 3%. The high contribution of $NO_2$ in comparison to NO and HONO, and the
existence of secondary pollutants, in particular PANs, also indicate that chemistry occurred from the time of
emission to the time of detection. Given the fast conversion of NO and HONO to $NO_2$ and nitrate, and $NH_3$ to
particulate ammonium, we also include in Table 3 the conserved quantity of $NO_y$, as well as $NO_x$ as NO, and $NH_x$
as $NH_3$ + particulate ammonium. Emissions of $SO_x$ as $SO_2$ that include the conversion of $SO_2$ to particulate sulfate
are discussed in Rickly et al. (2022).

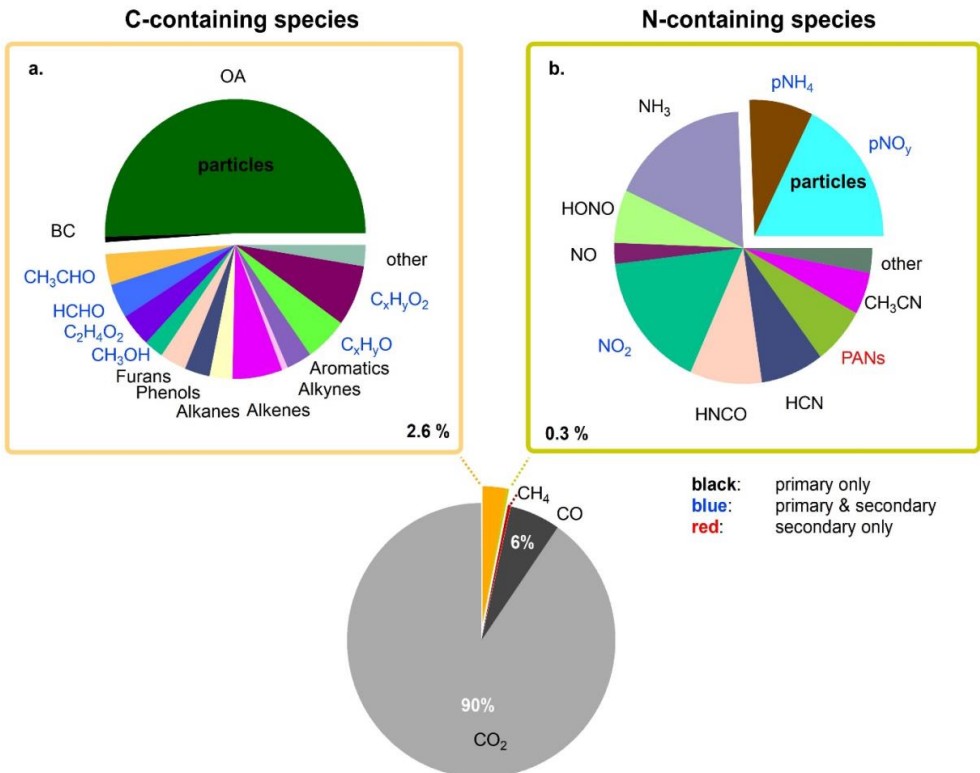

**Figure 3:** Pie charts of carbon- and nitrogen-containing species average emission factors (g kg$^{-1}$) for fresh wildfire smoke. The
text labels indicate compounds with only direct emissions in black, and compounds that are directly emitted and photochemically
produced in blue, and PANs that are only photochemically produced in red, indicating some oxidation even for the freshest
plumes sampled. Although HCHO and $CH_3CHO$ are $C_xH_yO$ species and glycolaldehyde/acetic acid are $C_xH_yO_2$ species they are
separately presented due to their high abundances.
**3.4 FIREX-AQ field observations compared to laboratory and field studies**
The sum of the NMOG EFs sampled during the FIREX-AQ campaign was 23.80 ± 7.5 g kg$^{-1}$ (3σ), in agreement
with the mean sum from western wildfires during the WE-CAN campaign of 26.1 ± 6.9 g kg$^{-1}$ (Permar et al., 2021),
temperate forest fires at 23.7 g kg$^{-1}$ (Akagi et al., 2011) and 24.55 g kg$^{-1}$ (Andreae, 2019), pine-forest understory
prescribed fires at 27.6 g kg$^{-1}$ (Yokelson et al., 2013), FLAME-4 laboratory coniferous canopy fires at 23.9 g kg$^{-1}$
(Stockwell et al., 2015), and FireLab laboratory measurements of various different fuel types at 25 g kg$^{-1}$ (Koss et





al., 2018). The sum of FIREX-AQ NMOG ERs to CO on a molar basis was $135 \pm 18$ ppb ppm$^{-1}$, in a similar range
as WE-CAN at $148.3 \pm 29.6$ ppb ppm$^{-1}$ and FireLab at $144.5$ ppb ppm$^{-1}$.

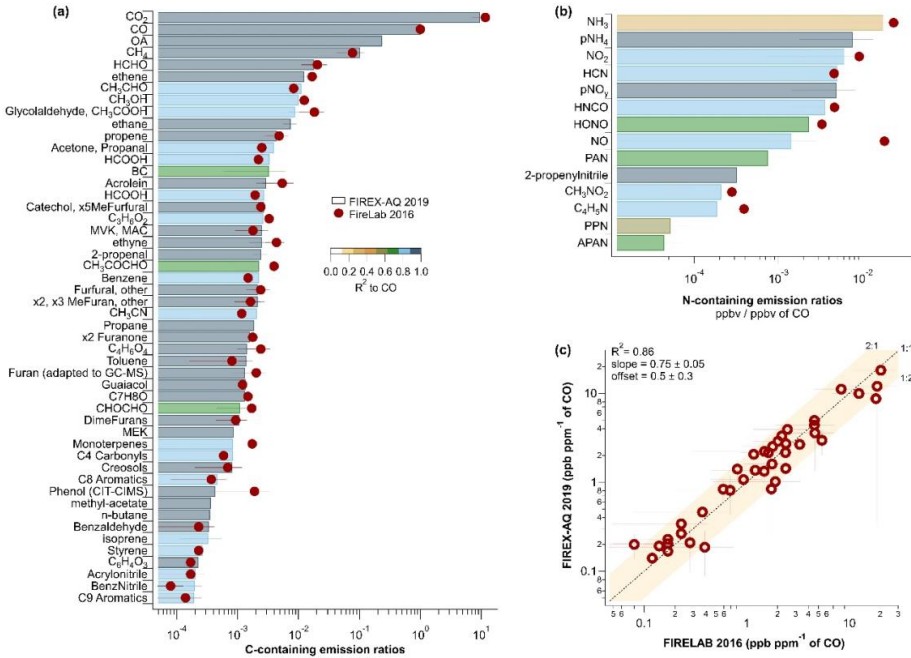


**Figure 4:** (a) and (b) show the emission ratios for FIREX-AQ (bars) and FireLab (circles) colored by the correlation coefficient,
and (c) direct comparison of FIREX-AQ to FireLab emission ratios for gas-phase species. Error bars in all graphs indicate the 1-
sigma standard deviation. The majority of the observations from FireLab 2016 were calculated using data from the NOAA PTR-
ToF-MS; here we use measurements from the same instrument for FIREX-AQ for more direct comparisons.
Figure 4 compares the ERs of C-containing and N-containing compounds (ppb ppb$^{-1}$ CO) with those measured at
the FireLab (Koss et al., 2018; Selimovic et al., 2018). During FIREX-AQ, all NMOGs correlated well with CO
with correlation coefficients, $R^2$, above 0.75, confirming that CO could be used as a proxy for estimating NMOG
emissions close to the fire, as further discussed in Sect. 3.5. Variability in the correlations of individual species with
CO was still evident — for example, species that are both emitted and photochemically produced exhibited lower
correlation (e.g., acetic acid, acetone, and formic acid, $R^2 = 0.75$–$0.85$) than compounds with only primary emissions
from fires (e.g., aromatics, $R^2 > 0.95$). N-containing species were weakly correlated with CO partly due to varying
fuel N/C (Roberts et al., 2020). In addition, lower correlation of $NH_3$ could be due to variable amounts of ammonium
formation in aging smoke, or differences in instrument response times between a high volatility compound, such as
CO, compared to $NH_3$, which may partition to the inlet and instrument walls before detection (Tomsche et al., 2023;
Stockwell et al., 2014) and slow the instrument response time. Low correlations are also found for HONO, which is
highly reactive and removed by photochemistry (Peng et al., 2020; Theys et al., 2020), as well as for glyoxal and
methylglyoxal, which are photochemically formed and could partition differently to the particle phase depending on
humidity (Mitsuishi et al., 2018; Ling et al., 2020). N-containing species were in good agreement except the higher
contribution of NO and particulate ammonium in Firelab and FIREX-AQ, respectively. This difference reflects the
depletion of NO and the secondary formation of particulate ammonium in field observations and promotes that fast
chemistry of reactive compounds occurred prior to the FIREX-AQ sampling. In summary, variability in post-
emission processes, fuel nitrogen, and fast photochemistry are likely important factors that contribute to the
differences in correlations between FIREX-AQ and Firelab measurements of NMOGs, $NO_y$ species, and CO.

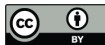



While the PTR-ToF-MS is well-suited for detecting NMOGs, it is prone to fragmentation for a range of molecules,
depending on their molecular structure (Pagonis et al., 2019). For such compounds, measurement uncertainties
increase, and comparisons to previous studies that use different instrumentation become more challenging. As
outlined in Sect. 2, the NOAA PTR-ToF-MS used in this study was the same instrument as used in the FireLab three
years prior (Koss et al., 2018). This provided an important opportunity to compare field-derived emissions to
laboratory studies. FireLab average ERs were calculated by comparing similar fuel types as measured during
FIREX-AQ, including Ponderosa Pine, Lodgepole Pine, Douglas Fir, Subalpine Fir, Engelmann Spruce, Loblolly
Pine, Jeffrey Pine, Juniper, Manzanita, Chamise, and Bear Grass laboratory fires. Overall, FIREX-AQ ERs agree
with those from the FireLab within a factor of 2 for most compounds (see Figure S3). Compounds with the largest
differences were benzonitrile with a FIREX-AQ to FireLab ratio of 2.46, ethene (1.88), $CH_3CN$ (1.77), toluene
(1.71), HCOOH (1.64), the sum of acetone and propanal (1.62), glycolaldehyde and acetic acid (0.50), monoterpenes
(0.49), $C_4H_5N$ species (0.47), syringol (0.32), and ethanol (0.28).

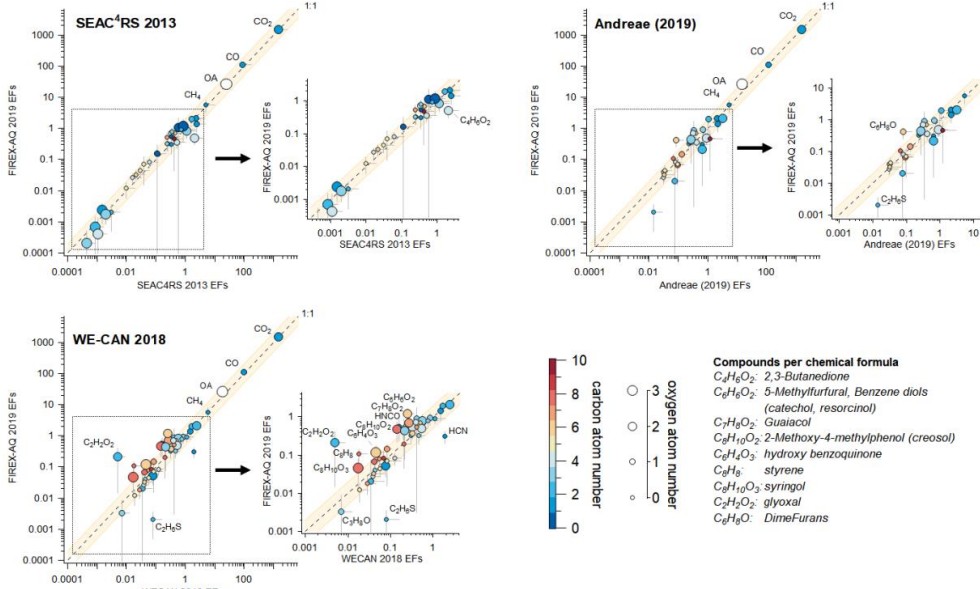


**Figure 5:** Comparison of FIREX-AQ EFs to those from SEAC⁴RS 2013 (Liu et al., 2017), WE-CAN (Permar et al., 2021), and
the review publication by Andreae (2019). Shaded areas show differences within a factor of 2.

Figure 5 and Table S6 compare FIREX-AQ observations against field-derived wildfire EFs from SEAC⁴RS (Liu et
al., 2017), WE-CAN (Permar et al., 2021), and literature-average temperate forest EFs from Andreae (2019). For all
studies, the measurements agree within a factor of 2 for 83%, 87%, and 78% of the compounds reported during
SEAC⁴RS, WE-CAN, and the Andreae (2019) temperate forest fires average (includes SEAC⁴RS), respectively.
FIREX-AQ EFs were on average higher compared to previous studies. The average ratio (± 1σ) of FIREX-AQ to
WE-CAN, SEAC⁴RS, and temperate forest fires from Andrea (2019) were $1.42 \pm 0.3$, $1.26 \pm 0.42$, and $1.24 \pm 0.36$,
respectively (see Table S6). Glyoxal and methylglyoxal were expected to have higher discrepancies due to their
secondary production and RH-dependent particle-phase partitioning, but also due to the higher quantification
uncertainties in the previous studies. For example, during WE-CAN (Permar et al., 2021), a PTR-ToF-MS was used
to detect these compounds, which are prone to fragmentation upon ionization in the PTR-ToF-MS. Furthermore, the
calculated glyoxal sensitivity used by Permar et al. (2021) was high (Stönner et al., 2016) and could therefore lead
to a significant underestimation. In this study, glyoxal and methylglyoxal were measured by cavity-enhanced
spectroscopy, and the uncertainties were < 5% (see Sect. 2). Furthermore, comparison of the FIREX-AQ to the
FireLab EFs also measured by the same spectroscopic technique (see Figure 4) (Zarzana et al., 2018) showed that
glyoxal and methylglyoxal were in better agreement with FIREX-AQ compared to Permar et al. (2021) but still



lower by 50% and 75%, respectively. Dimethyl sulfide (DMS) is a compound that originates predominantly from
oceanic emissions and its fire emissions were lower for this study compared to WE-CAN and the temperate forest
fire emissions average, but higher by 20% compared to the SEAC4RS EFs. FIREX-AQ monoterpenes were higher
than those in WE-CAN and Firelab by a factor of 2, and lower than the temperate forest fire emissions average
(Andreae 2019) by a factor of 2, which likely stems from the large variability of monoterpene emissions for different
fuel types and the difficulties inherent with the large number of isomers (Hatch et al. 2017; Koss et al. 2018;
Sekimoto et al. 2018). OA was 50% higher compared to WE-CAN and temperate forest fire emissions, but within
10% when compared to the SEAC$^4$RS OA emissions. The variability of OA EFs highlights the importance of
accounting for the partitioning and aging of OA when comparing OA EFs across biomass burning campaigns given
that fraction of the detected OA from wildfire plumes can be a mix of primary and secondary (Pagonis et al., 2020).
Focusing on the two large recent campaigns dedicated to wildfires we note that differences can occur due to natural
variability with 2018 being a more intense fire season (Jin et al., 2023), but also from the different fragmentation,
inlet setups, and quantification uncertainties between the instruments used. Differences between the WE-CAN and
FIREX-AQ EFs for oxygenated compounds could be due to the different quantification uncertainties between the
two PTR-ToF-MS instruments. For both studies and instruments, assuming similar isomer sensitivities and no
fragmentation interferences, sensitivities for calibrated compounds introduced a 15% uncertainty, whereas
sensitivities for uncalibrated species were estimated following theoretical methods described by Sekimoto et al.
(2017), which have an uncertainty of 50%. Several reactive oxygenated compounds that have implications for $NO_x$
loss processes such as the formation of nitrophenolic compounds (Finewax et al., 2018; Decker et al., 2021) were
calibrated during FIREX-AQ but only calculated during WE-CAN, such as $C_7H_8O$ (o-cresol, anisol), $C_7H_8O_2$
(guaiacol), and $C_8H_{10}O_2$ (creosol). One mass calibrated on both instruments was $C_6H_6O_2$ (sum of 5-methyl furfural,
catechol, resorcinol), but was still a factor of 5 higher during FIREX-AQ compared to WE-CAN. However, the
FIREX-AQ ERs for $C_6H_6O_2$ agreed within 45% of the FireLab study, which used the same instrument, suggesting
possible differences in fragmentation or isomer assignment between the FIREX-AQ and WE-CAN instruments.
Styrene ($C_8H_8$) from FIREX-AQ (using PTR-MS) was a factor of 6 higher compared to the WE-CAN measurements
(GC-MS) but agreed within 60% with SEAC$^4$RS (GC-MS) and FireLab EFs (PTR-MS). $C_6H_8O$ (sum of 2,5-
dimethylfuran, 2-ethylfuran, and other $C_2$-substituted furan isomers), $C_8H_{10}O_3$ (syringol), and $C_6H_4O_3$ (hydroxy
benzoquinone) were quantified using estimated calibration factors during both campaigns, and therefore more
uncertain, and were higher by a factor of 2–5 during FIREX-AQ. Another influencing factor for the overall higher
EFs for oxygenated compounds during FIREX-AQ could be due to the optimized inlet setups to limit wall losses
prior to detection for the majority of the instruments (Table 2). Various oxygenated compounds are more sticky and
can therefore partition to the inlet line walls prior to their detection. For example, during FIREX-AQ the NOAA
PTR-ToF-MS inlet line was 1-m long and heated at 60°C to reduce condensation sinks resulting in less than 1 s
residence times; in Firelab (Koss et al., 2018) a longer 16 m transfer line was used at 40°C with a residence time
comparable to FIREX-AQ whereas in WE-CAN (Permar et al., 2021) the smoke to drift tube time was higher (~ 2
s) at temperatures of 55-60°C. This could therefore contribute to differences for larger or more oxygenated NMOGs
between campaigns and partly explain the overall increased EFs during FIREX-AQ.
Further differences between FIREX-AQ and WE-CAN may also result from the methods used to identify and
characterize young plumes. As described in Section 3.1, fresh plumes are identified during FIREX-AQ based on
chemical aging proxies, whereas fresh plumes identified in WE-CAN are based on physical distance downwind. For
highly reactive species, such as furans and oxygenated aromatics, strong fire-to-fire variability in OH exposure may
alter emission factors, even in smoke with similar downwind age. Figure S2 compares the FIREX-AQ and WE-
CAN field observations to the ERs obtained during the FireLab laboratory study for a variety of overlapping NMOGs
with varying reactivities towards OH radicals. Given that FireLab experiments were performed under dark and
warmer conditions in smoke aged just 5 s, it is expected that the more reactive compounds would show higher ERs
when compared to field observations if the sampled smoke onboard the aircraft was already aged. However, higher
ERs were observed for various compounds measured during FIREX-AQ. On the contrary, when comparing WE-
CAN to FireLab ERs, the highly reactive compounds were lower although the ERs of less reactive compounds were
in good agreement. This indicates possible differences between FIREX-AQ and WE-CAN owing to variability in
chemical oxidation, which has the largest impact on highly reactive species.



The correlation to MCE for each species EFs was calculated for all wildfires as shown in Table S4 and compared to
the WE-CAN observations. Correlation coefficients ($R^2$) during FIREX-AQ were above 0.5 for 28% of the species,
0.3–0.5 for 27% of the species, and below 0.3 for the remaining species. The lowest correlations, below 0.1, were
found for N-containing species, including particulate ammonium and pNOy, ammonia, acetonitrile, 2-butyl nitrate,
methyl nitrate, pyrrole and butene nitrile isomers, and acrylonitrile. Nevertheless, agreement within a factor of 2
was found when compared to the slopes and $R^2$ obtained from the WE-CAN campaign for most of the compounds.
Figure 6 shows the dependence of two N-containing species on fire MCEs for the FIREX-AQ and FireLab (Roberts
et al., 2020) studies as well as for a majority of fuel types by Akagi et al. (2011) and Andreae (2019). We report N-
containing species as a ratio to the total reactive nitrogen $N_r$, defined as the sum of NO, $NO_2$, HONO, HNCO, HCN,
$NH_3$, other N-containing VOCs, and particle-phase nitrate and ammonium. The dotted lines and shaded regions
show FireLab parameterizations that describe how these ratios respond to changes in MCE (Roberts et al., 2020) for
one fire burned during FireLab whereas square and bended square markers indicate different land cover types from
Andreae (2019) and Akagi et al. (2011), respectively. It should be noted that for Akagi et al. (2011) and Andreae
(2019) $N_r$ measurements are limited to the sum of NO, $NO_2$, HONO, HCN, and $NH_3$ and therefore the $N_r$ could
represent a lower limit. For both laboratory and field studies and independent of the fuel burnt, as MCE increases,
$NO_x/N_r$ increases, whereas $NH_3/N_r$ decreases. The FireLab MCE ranged from pure flaming (MCE = 0.99) to
smoldering values (MCE<0.8), but ambient observations during FIREX-AQ were limited to MCE values ranging
from 0.85 to 0.95, which suggests both flaming and smoldering contributions to the sampled wildfire plumes.

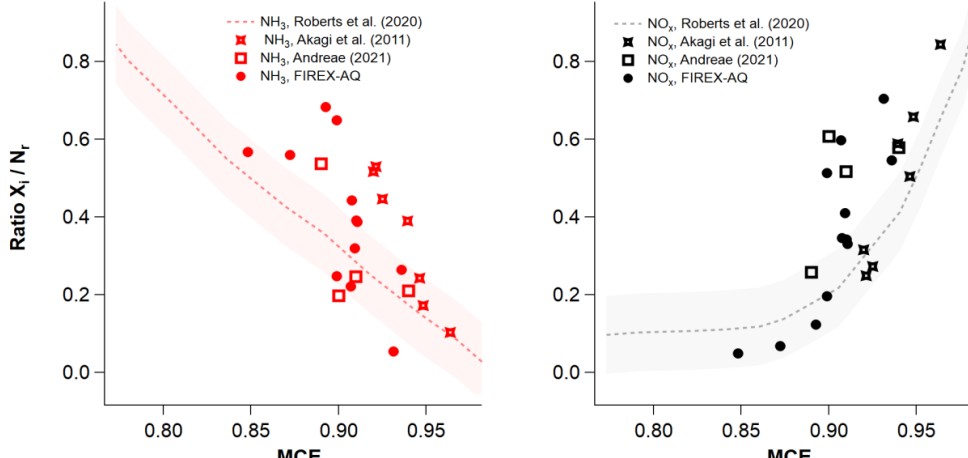


**Figure 6:** Ratios of two N-species to the total nitrogen, $N_r$, during FIREX-AQ compared to Roberts et al. (2020) based on one
fire burned during FireLab, and Andreae (2019) and Akagi et al. (2011) that include different land cover types.
**3.5 Parameterization of organic- and nitrogen-containing emissions in wildfire plumes**
The comparisons described above demonstrate that FIREX-AQ emissions agreed within a factor of 2 or better with
previous laboratory and field studies for most C- and N-containing species for temperate ecosystem fuels. In the
following, we relate primary wildfire emissions and emission factors to fire emissions measurable from space, e.g.,
CO (e.g., Schneising et al., 2020), $NO_2$ (e.g., Martínez-Alonso et al., 2020), and BC (e.g., Konovalov et al., 2018),
as well as MCE. Although current satellite retrievals for wildfire smoke can agree with airborne observations e.g.,
for $NO_x$ and CO(Griffin et al., 2021; Stockwell et al. 2022), challenges in isolating the fire contribution from small
or short-lived fires, as well as cloud coverage and aerosol interferences, add uncertainties to this quantification (e.g.,
Jung et al., 2019; Vasilkov et al., 2021). Here, we only focus on the parameterization of wildfire plumes and promote
future efforts to quantify these compounds using satellite retrievals more accurately. Satellite-retrieved
concentrations of CO and $NO_2$ close to wildfires could then be used to estimate NMOG and $NO_y$ emissions and
potentially better account for variability associated with fire emissions and improve modeling efforts to simplify
and predict downwind formation of secondary pollutants, including ozone and secondary organic aerosol.





Figure 7 shows correlations between the sum of the median mixing ratios of NMOGs and $NO_y$ with MCE, CO, and
$NO_2$, where CO and $NO_2$ are two species available from satellite products that could be used as proxies for
smoldering and flaming combustion (e.g., van der Velde et al., 2021; Urbanski et al., 2008), respectively. Figure 7a
shows that the sum of FIREX-AQ NMOG EFs correlated with MCE with an $R^2$ of 0.68, even though many of the
individual compounds are poorly correlated with MCE (Table S4). The correlation of the FIREX-AQ MCE to the
sum of NMOGs was in the same range as WE-CAN, FireLab, and FLAME-4 observations. WE-CAN was
consistently lower, by around 10%, which is partially due to differences in the assumed fraction of carbon employed
in Eq. (1) (45.7% for WE-CAN and 50% for this study). FIREX-AQ sampled fires with lower MCEs on average
than the lab experiments, with lab experiments showing highly variable EFs for MCE values below 0.9. Additional
reasons for different FireLab and FLAME-4 EFs vs. MCE are discussed in detail by Permar et al. (2021) and include,
(1) rapid chemistry prior to sampling, which results in the degradation of short-lived species (Figure S2) and/or less
partitioning to particles at higher lab temperatures, (2) laboratory studies may more efficiently sample smoldering
combustion emissions compared to aircraft observations where residual smoldering combustion emissions might
not be lofted and therefore undersampled at the aircraft altitude, and (3) laboratory MCEs are often higher than in
the field due to experimental conditions, including drier fuel and more efficient burning conditions (Yokelson et al.,
2013; Holder et al., 2017; Selimovic et al., 2018), whereas field MCEs are calculated from single transects through
smoke plumes that likely contain a different mix of flaming vs. smoldering (Wiggins et al., 2020). Nevertheless, the
good agreement between two different aircraft studies during different years and the general agreement with FireLab
and FLAME-4 study averages further highlight the consistency of total NMOG correlations with MCE in wildfire
emissions despite the poorer correlations of individual compounds with MCE (Table S4).

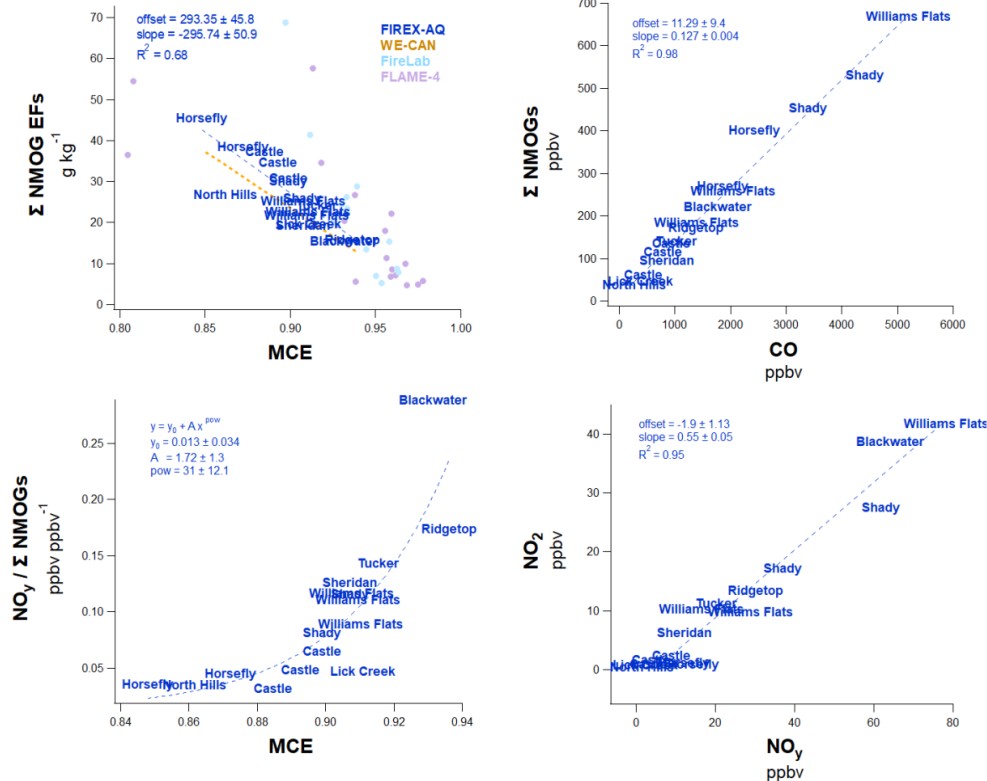


**Figure 7**: Correlation trends observed for western US wildfire emissions for (a) the sum of median NMOG EFs compared to
MCE for each wildfire. Each data point represents one fire from either FIREX-AQ, WE-CAN (Permar et al., 2021), FireLab
(Koss et al., 2018), or FLAME-4 (Stockwell et al., 2015) with the name of each FIREX-AQ fire centered on the data points. (b)





Sum of median NMOG mixing ratios plotted vs. CO, (c) ratio of median NOy species to the sum of NMOGs vs. MCE, and (d)
median NOy mixing ratios vs. the median NO2 concentration. Dashed lines indicate linear fits for (a), (b), and (d), and a power
function fit for (c).
Figure 7b relates the sum of the median NMOG mixing ratios to the median CO mixing ratios for all the freshest
sampled wildfire plumes. CO results largely from smoldering combustion, which is the combustion process that also
produces most NMOGs. NMOGs and CO are very well correlated, with a slope of $127 \pm 4$ (ppb ppm$^{-1}$) and an R$^2$
of 0.98, which demonstrates that total primary NMOG emissions are effectively represented by CO. Figure S4 shows
that R$^2$ values with CO for individual compounds were above 0.9 for the majority of primary NMOGs reported here,
whereas, for secondary species, the correlations were below 0.3. CO columns are retrievable from space by, e.g.,
TROPOMI (Martínez-Alonso et al., 2020) and CRiS (NASA, 2015) and can be used to derive CO emissions that
generally agree with *in situ* observations (Stockwell et al. 2022). The correlations from the FIREX-AQ
measurements and others could be used to initialize total NMOG emissions from wildfire plumes in models.
Quantification of N-containing species is also essential for understanding and modeling the evolution and formation
of secondary organic aerosol and ozone downwind of wildfires. Figure 7c shows the ratio of measured NOy by the
chemiluminescence instrument (see Section 2.1), to the sum of NMOGs in ppb ppb$^{-1}$. A rapid increase in this ratio
is observed as MCE increases described by a power function fit. This increase follows the expectation that as fires
transition from smoldering to flaming conditions, MCE increases, NMOGs EFs decrease, and fuel nitrogen leads to
the formation of NO$_x$ through radical chemistry of N-containing compounds (Roberts et al., 2020). Figure 7d shows
that NO$_2$ represents a significant fraction of NOy with a slope of $0.55 \pm 0.03$ (ppb ppb$^{-1}$) and an R$^2$ of 0.95.
Furthermore, the correlation of individual N-containing species with NO$_2$ is significantly higher than their
correlation with CO mixing ratios (Figure S4) promoting that NO$_2$ measurements could be used to initialize total
NO$_y$ emissions and N-species from wildfire plumes in models. Figures S5 shows additional correlations that could
be used for modeling efforts, including the correlation of NO$_y$ to CO, NO$_y$ to BC, and others.
These observations suggest that CO is a good proxy for species emitted from western wildfires primarily during
smoldering conditions (i.e., NMOGs), whereas NO$_2$ is a good proxy for species that are mostly emitted during
flaming conditions (i.e., mostly NO$_y$). Thus, in addition to coupling EFs with fuel consumption to derive emissions,
we suggest future use of satellite retrievals close to the fire plume to quantify CO and NO$_2$ concentrations in order
to accurately determine EFs for all carbon and nitrogen-containing species for western US wildfire plumes as input
to models. An important assumption, especially in determining emissions of N-containing species, is that NO$_2$
should accurately represent NO$_y$ close to the fire. However, satellite retrievals that capture truly fresh emissions very
close to the fire will be dominated by NO and HONO whereas in highly oxidized plumes NO$_2$ loss processes will
lower its overall contribution to NO$_y$. It is therefore important to provide a range of distances where this holds true.
Coggon et al. (2022) find that for fires with highly reactive emissions, NO$_2$ represents NO$_y$ within the first 15-30
min and a distance of 10-20 km downwind of the fire assuming a wind speed 10 m/s. Current satellite retrievals for
wildfire smoke have a spatial resolution of 3.5 km×5.5 km (Griffin et al., 2021) which would be within the above
range and high enough to represent plumes where NO$_2$ is the dominant fraction of NO$_y$.
**Table 3:** Emission ratios and emission factors of organic and nitrogen compounds from wildfire plumes. In blue are multiple
isomers measured as sum by the NOAA PTR-ToF-MS that were further speciated based on other GC-MS measurements from
FIREX-AQ (column 1 in parenthesis). Here, we show the ratio of each isomer measured by GC-MS to the total PTR-ToF-MS
signal obtained in this mass.

| Compound<br>*Isomer contribution to each mass is provided in parenthesis based on the ratio of each isomer measured by GC-MS to the sum measured by PTR-ToF-MS (check Table S5)* | Instrument | Exact Mass, Da | Chemical formula/ structure | EFs (g kg$^{-1}$) | ± σ | ERs (ppb ppm$^{-1}$) | ± σ |
|---|---|---|---|---|---|---|---|
| **Gas-Phase** | | | | | | | |
| Carbon dioxide | DACOM | 43.99 | CO2 | 1533.82 | 78.06 | 9400.32 | 2455.30 |



| Carbon monoxide | DACOM | 27.99 | CO | 109.15 | 22.70 | 1000.00 | 0.00 |
|---|---|---|---|---|---|---|---|
| Methane | DACOM | 16.03 | CH4 | 5.81 | 2.68 | 91.97 | 31.61 |
| Formaldehyde | CAMS & ISAF | 30.01 | CH2O | 2.10 | 0.79 | 17.92 | 4.31 |
| Acetic acid + Glycolaldehyde | NOAA PTR-ToF-MS for the sum | 60.02 | C2H4O2 | 2.09 | 0.61 | 8.86 | 1.51 |
| Acetaldehyde | NOAA PTR-ToF-MS | 44.03 | C2H4O | 1.95 | 0.60 | 11.25 | 1.70 |
| Ethene | iWAS | 28.03 | C2H4 | 1.52 | 0.45 | 13.57 | 1.97 |
| Methanol | NOAA PTR-ToF-MS | 32.03 | CH4O | 1.42 | 0.66 | 10.90 | 3.21 |
| 5-Methylfurfural + Benzene diols (=Catechol, Resorcinol) | NOAA PTR-ToF-MS for the sum | 110.11 | C6H6O2 | 1.20 | 0.47 | 2.72 | 0.68 |
| Acetone (78%) + Propanal (22%) | NOAA PTR-ToF-MS (speciation by GC-MS) | 58.04 | C3H6O | 0.93 | 0.34 | 4.04 | 0.84 |
| Ethane | iWAS | 30.05 | C2H6 | 0.91 | 0.26 | 7.76 | 1.84 |
| Methyl acetate + Ethyl formate + Hydroxyacetone | NOAA PTR-ToF-MS for the sum | 74.04 | C3H6O2 | 0.81 | 0.36 | 2.70 | 0.73 |
| Propene | iWAS | 42.05 | C3H6 | 0.80 | 0.27 | 4.80 | 1.16 |
| MVK (38%) + Methacrolein (27%) + 2-Butenal (33%) | NOAA PTR-ToF-MS (speciation by GC-MS) | 70.09 | C4H6O | 0.71 | 0.27 | 2.56 | 0.56 |
| Benzene | NOAA PTR-ToF-MS | 78.05 | C6H6 | 0.69 | 0.17 | 2.26 | 0.24 |
| Guaiacol (=2-Methoxyphenol) | NOAA PTR-ToF-MS | 124.14 | C7H8O2 | 0.70 | 0.34 | 1.38 | 0.52 |
| Acrolein | NOAA PTR-ToF-MS | 56.03 | C3H4O | 0.88 | 0.88 | 3.73 | 2.73 |
| Methyl glyoxal | ACES | 72.06 | CH3COCHO | 0.44 | 0.36 | 1.55 | 1.23 |
| Isocyanic acid | NOAA PTR-ToF-MS | 43.01 | HNCO | 0.53 | 0.31 | 3.51 | 2.46 |
| Formic acid | NOAA PTR-ToF-MS | 46.00 | HCOOH | 0.60 | 0.43 | 3.31 | 1.95 |
| 2-Methylphenol (=o-cresol) + Anisol | NOAA PTR-ToF-MS for the sum | 108.14 | C7H8O | 0.57 | 0.22 | 1.32 | 0.37 |
| 2-(3H)-Furanone | NOAA PTR-ToF-MS | 84.02 | C4H4O2 | 0.54 | 0.26 | 1.60 | 0.50 |
| HCN | CIT-CIMS | 27.01 | HCN | 0.31 | 0.12 | 3.01 | 1.08 |
| Toluene | NOAA PTR-ToF-MS | 92.06 | C7H8 | 0.53 | 0.21 | 1.42 | 0.35 |
| 2,3-Butanedione + 2-Oxobutanal + 1,4-Butanedial | NOAA PTR-ToF-MS for the sum | 86.04 | C4H6O2 | 0.49 | 0.20 | 1.43 | 0.37 |
| Monoterpenes | NOAA PTR-ToF-MS | 136.24 | C10H16 | 0.47 | 0.43 | 0.82 | 0.65 |
| 2-Methoxy-4-methylphenol (= Creosol) | NOAA PTR-ToF-MS | 138.16 | C8H10O2 | 0.47 | 0.26 | 0.82 | 0.36 |
| 2,5-Dimethylfuran + 2-Ethylfuran + Other unidentified organic compounds | NOAA PTR-ToF-MS for the sum | 96.06 | C6H8O | 0.41 | 0.16 | 1.07 | 0.27 |
| Phenol | CIT-CIMS | 94.04 | C6H6O | 0.16 | 0.05 | 0.43 | 0.13 |
| Furan | TOGA | 68.03 | C4H4O | 0.35 | 0.13 | 1.33 | 0.40 |
| i-Butene | iWAS | 56.06 | C4H8 | 0.35 | 0.12 | 1.61 | 0.42 |
| Acetonitrile | NOAA PTR-ToF-MS | 41.03 | C2H3N | 0.32 | 0.14 | 2.04 | 0.86 |
| Propane | iWAS | 44.06 | C3H8 | 0.33 | 0.14 | 1.90 | 0.66 |
| Ethyne | iWAS | 26.02 | C2H2 | 0.30 | 0.14 | 2.90 | 0.92 |
| Glyoxal | ACES | 58.04 | CHOCHO | 0.22 | 0.20 | 0.94 | 0.78 |
| MEK | NOAA PTR-ToF-MS | 72.06 | C4H8O | 0.24 | 0.08 | 0.84 | 0.20 |
| Ethylbenzene (7%) + m- and p-Xylenes (58%) + o-Xylene (21%) | NOAA PTR-ToF-MS (speciation by GC-MS) | 106.17 | C8H10 | 0.08 | 0.04 | 0.18 | 0.07 |



| 2-Furfural | TOGA | 96.02 | C5H4O2 | 0.18 | 0.06 | 0.47 | 0.11 |
|---|---|---|---|---|---|---|---|
| Benzaldehyde | NOAA PTR-ToF-MS | 106.12 | C7H6O | 0.15 | 0.05 | 0.35 | 0.06 |
| Butene | iWAS | 56.06 | C4H8 | 0.15 | 0.05 | 0.68 | 0.16 |
| Hydroxy benzoquinone | NOAA PTR-ToF-MS | 124.09 | C6H4O3 | 0.12 | 0.06 | 0.23 | 0.09 |
| 2-Methylfuran | TOGA | 82.04 | C5H6O | 0.11 | 0.04 | 0.34 | 0.10 |
| Styrene | NOAA PTR-ToF-MS | 104.15 | C8H8 | 0.11 | 0.04 | 0.26 | 0.06 |
| C9 Aromatics | NOAA PTR-ToF-MS | 120.19 | C9H12 | 0.084 | 0.043 | 0.178 | 0.073 |
| Naphthalene | NOAA PTR-ToF-MS | 128.17 | C10H8 | 0.077 | 0.032 | 0.161 | 0.074 |
| n-Butane | iWAS | 58.08 | C4H10 | 0.082 | 0.030 | 0.368 | 0.121 |
| Benzonitrile | NOAA PTR-ToF-MS | 103.04 | C7H5N | 0.081 | 0.027 | 0.200 | 0.062 |
| Pentene | iWAS | 70.08 | C5H10 | 0.073 | 0.023 | 0.268 | 0.069 |
| Benzofuran | NOAA PTR-ToF-MS | 118.10 | C8H6O | 0.067 | 0.023 | 0.143 | 0.031 |
| Butanal | TOGA | 72.06 | C4H8O | 0.060 | 0.019 | 0.217 | 0.064 |
| Isoprene | iWAS | 68.06 | C5H8 | 0.070 | 0.055 | 0.271 | 0.203 |
| Propyne | WAS | 40.03 | C3H4 | 0.057 | 0.027 | 0.362 | 0.121 |
| 2-Methyl-1-butene | iWAS | 70.08 | C5H10 | 0.055 | 0.020 | 0.201 | 0.054 |
| Nitromethane | NOAA PTR-ToF-MS | 61.02 | CH3NO2 | 0.052 | 0.025 | 0.228 | 0.116 |
| 1-Hexene | WAS | 84.09 | C6H12 | 0.049 | 0.013 | 0.151 | 0.043 |
| 2-Methylpropanal | TOGA | 72.06 | C4H8O | 0.046 | 0.015 | 0.167 | 0.049 |
| n-Pentane | iWAS | 72.09 | C5H12 | 0.044 | 0.018 | 0.159 | 0.058 |
| Acrylonitrile | NOAA PTR-ToF-MS | 53.03 | C3H3N | 0.040 | 0.011 | 0.202 | 0.073 |
| Cis-2-Butene | iWAS | 56.06 | C4H8 | 0.013 | 0.005 | 0.045 | 0.013 |
| 2-Methyl-1-Butene | iWAS | 70.08 | C5H10 | 0.055 | 0.020 | 0.201 | 0.054 |
| Syringol | NOAA PTR-ToF-MS | 154.17 | C8H10O3 | 0.047 | 0.034 | 0.078 | 0.056 |
| Pentadiene | iWAS | 68.06 | C5H8 | 0.033 | 0.015 | 0.123 | 0.044 |
| Trans-2-Butene | iWAS | 56.06 | C4H8 | 0.037 | 0.020 | 0.166 | 0.082 |
| n-Hexane | iWAS | 86.11 | C6H14 | 0.033 | 0.013 | 0.099 | 0.038 |
| i-Butane | iWAS | 58.08 | C4H10 | 0.027 | 0.010 | 0.122 | 0.038 |
| 1-Heptene | WAS | 98.11 | C7H14 | 0.026 | 0.008 | 0.069 | 0.022 |
| Ethanol | NOAA PTR-ToF-MS | 46.04 | C2H6O | 0.020 | 0.055 | 0.098 | 0.273 |
| n-Nonane | iWAS | 128.16 | C9H20 | 0.025 | 0.010 | 0.051 | 0.020 |
| Methyl Formate | iWAS | 60.02 | C2H4O2 | 0.020 | 0.022 | 0.089 | 0.095 |
| n-Decane | iWAS | 142.17 | C10H22 | 0.023 | 0.012 | 0.042 | 0.024 |
| 3-Methylfuran | TOGA | 82.04 | C5H6O | 0.019 | 0.006 | 0.058 | 0.017 |
| 1-Octene | WAS | 112.13 | C8H16 | 0.018 | 0.005 | 0.042 | 0.013 |
| 3-Furfural | TOGA | 96.02 | C5H4O2 | 0.018 | 0.006 | 0.047 | 0.011 |
| Trans-2-Pentene | iWAS | 70.08 | C5H10 | 0.018 | 0.008 | 0.065 | 0.025 |
| 2,4-Dimethylpentane | iWAS | 100.13 | C7H16 | 0.018 | 0.009 | 0.046 | 0.019 |
| 1-Nonene | WAS | 126.14 | C9H18 | 0.015 | 0.005 | 0.031 | 0.011 |
| 1-Buten-3-yne | WAS | 52.03 | C4H4 | 0.014 | 0.007 | 0.070 | 0.026 |
| Pyrrole | TOGA | 67.04 | C4H5N | 0.012 | 0.005 | 0.047 | 0.024 |
| i-Pentane | iWAS | 72.09 | C5H12 | 0.012 | 0.006 | 0.045 | 0.023 |
| cis-2-Butene | iWAS | 70.08 | C5H10 | 0.013 | 0.005 | 0.045 | 0.013 |
| Butene nitrile isomers | TOGA | 67.04 | C4H5N | 0.007 | 0.003 | 0.028 | 0.014 |
| 2-Methylpentane | iWAS | 86.11 | C6H14 | 0.007 | 0.003 | 0.020 | 0.008 |
| 1-Butyne | WAS | 54.05 | C4H6 | 0.006 | 0.003 | 0.030 | 0.012 |



| | | | | | | | |
|---|---|---|---|---|---|---|---|
| Methylcyclopentane | iWAS | 84.09 | C6H12 | 0.005 | 0.002 | 0.015 | 0.006 |
| Methylcyclohexane | iWAS | 98.11 | C7H14 | 0.004 | 0.002 | 0.011 | 0.006 |
| Dimethyl sulfide (50%) + other unidentified organic compounds (50%) | NOAA PTR-ToF-MS (speciation by GC-MS) | 62.02 | C2H6S | 0.002 | 0.002 | 0.009 | 0.007 |
| 2-Butyne | WAS | 54.05 | C4H6 | 0.003 | 0.002 | 0.014 | 0.008 |
| Methyl Nitrate | iWAS | 77.01 | CH3NO3 | 0.002 | 0.002 | 0.008 | 0.005 |
| i-Propanol | iWAS | 60.06 | C3H8O | 0.003 | 0.006 | 0.015 | 0.026 |
| i-Propyl nitrate | iWAS | 105.04 | C3H7NO3 | 0.002 | 0.001 | 0.005 | 0.002 |
| 1,3-Butadiyne | WAS | 50.02 | C4H2 | 0.001 | 0.001 | 0.006 | 0.002 |
| Ethyl nitrate | iWAS | 91.03 | C2H5NO3 | 0.001 | 0.001 | 0.002 | 0.002 |
| 2-Butyl nitrate | iWAS | 119.06 | C4H9NO3 | 0.0005 | 0.001 | 0.001 | 0.002 |
| | | | | | | | |
| NOy | CL | | NOy | | | 12.10 | 7.38 |
| Nitrogen dioxide | CL | 46.01 | NO2 | 1.09 | 1.22 | 6.05 | 5.34 |
| Nitric oxide | CL | 30.01 | NO | 0.16 | 0.16 | 1.42 | 1.44 |
| Nitrous acid | NOAA CIMS | 47.00 | HONO | 0.30 | 0.21 | 1.89 | 1.61 |
| Ammonia | Oslo PTR-ToF-MS | 17.03 | NH3 | 1.15 | 0.77 | 17.44 | 11.65 |

**Aerosol-Phase** *(all units in g/kg)*

| | | | | | | | |
|---|---|---|---|---|---|---|---|
| Organic aerosol (OA/OC = 1.85 ± 0.16) | AMS | | OA | 26.51 | 13.97 | 317.3 | 148.9 |
| Particulate nitrate | AMS | 62.00 | pNO$_y$ | 1.19 | 0.78 | 7.29 | 2.69 |
| Particulate ammonium | AMS | 18.04 | pNH4$^+$ | 0.53 | 0.37 | 3.24 | 1.97 |
| Black carbon | SP2 | | BC | 0.35 | 0.32 | 3.26 | 2.69 |

**Sums**

| | | | | | | | |
|---|---|---|---|---|---|---|---|
| | | | | NH3 EFs also derived in Tomsche et al. (2023) | | | |
| NHx as NH3 (EF$_{NH3}$ + (17/18)* EF$_{NH4}$) | UIBK/UiO PTR-ToF-MS + AMS | 17.03 | NH3 | 1.65 | 1.14 | 24.56 | 17.10 |
| NOx as NO (EF$_{NO}$ + (30/46)* EF$_{NO2}$) | CL | 30.01 | NO | 0.87 | 0.96 | 5.37 | 4.92 |
| SOx as SO2 | NO-LIF, AMS | See Rickly et al. (2022) | | | | | |
| | | **Total NMOGs emissions** | | 23.80 | 7.5 | 135.02 | 18.23 |


**Conclusions**

We present ERs and EFs for NMOGs and nitrogen-containing compounds from nine western US wildfires and one
southeastern US prescribed fire derived from data obtained aboard the NASA DC-8 during the 2019 FIREX-AQ
mission. ERs and EFs were calculated for a total of 16 crosswind plume transects chosen to represent the freshest
fire emissions. These transects were identified based on proxies (e.g., maleic anhydride/furan ratio) for chemical
aging, which can be rapid in fire plumes.
We performed detailed comparisons of FIREX-AQ emissions to previous laboratory and field studies with a focus
on oxygenated organic compounds that were calibrated during this mission. FIREX-AQ ERs agree within a factor
of 2 to the FireLab study for most compounds, with a correlation slope of 0.75 ± 0.05 and an $R^2$ of 0.86. A
comparison of the field-derived EFs from FIREX-AQ with those from SEAC$^4$RS (Liu et al., 2017), WE-CAN
(Permar et al., 2021), and temperate forest ERs from Andreae (2019) also agreed to within a factor of 2 for 87%,
83%, and 78% of the compounds, respectively. However, FIREX-AQ EFs are on average higher compared to



previous studies. For compounds that agree within a factor of 2, the average ratios of FIREX-AQ to WE-CAN, SEAC4RS, and the temperate forest fire literature average are $1.09 \pm 0.3$, $1.25 \pm 0.33$, and $1.18 \pm 0.4$, respectively, whereas for the remaining compounds, the ratios increase to $2.1 \pm 1.64$, $1.29 \pm 1.01$, and $1.32 \pm 1.23$. We suggest that these differences could be due to differences in the fuel, quantification methods applied for each study, as well as due to differences in photochemical loss of reactive species prior to detection. We further compare the ratio of N-containing species to the total nitrogen $(N_i/N_r)$ vs. MCE and find that $NO_x/N_r$ and $NH_3/N_r$ follow similar trends as those reported by Roberts et al. (2020).

We relate wildfire emissions of C- and N-containing species to CO, $NO_2$, BC, and MCE based on correlations for use in chemical transport models. Results show that the sum of NMOG EFs correlates with MCE, with an $R^2$ of 0.68 and a slope of $-296 \pm 51$ g $kg^{-1}$. A better correlation is observed between the sum of the median NMOG mixing ratios and median CO, with a slope of $0.127 \pm 0.004$ (ppb ppm-1) and an $R^2$ of 0.98. Consistent correlation of individual NMOGs to CO is also evident for the majority of NMOGs with $R^2$ values greater than 0.9, suggesting significant potential for estimating wildfire NMOG emissions using space-based CO emissions.

For N-containing species, the sum of reactive nitrogen, $NO_y$, correlates better with $NO_2$ ($R^2 = 0.95$, slope = $1.74 \pm 0.1$ ppbv $ppbv^{-1}$) and BC ($R^2 = 0.88$) than with CO ($R^2 = 0.7$) close to the wildfire. Furthermore, the ratio of $NO_y$ to the sum of NMOGs increases exponentially as MCE increases. This further highlights the important influence of fire behavior e.g., flaming vs. smoldering fire conditions on the emissions of reactive nitrogen species. Future efforts to initialize models using the above emissions parameterization could improve the representation of fire emissions in models and their predictions on the downwind formation of secondary pollutants like ozone and secondary organic aerosol.

## Acknowledgments

We would like to thank the NOAA/NASA FIREX-AQ science and aircraft operation teams. GIG, MMC, CES, MMB, IB, JMK, AL, SAM, JAN, JP, PSR, MAR, RHS, CCW, and CW were supported by the NOAA Cooperative Agreement with CIRES, NA17OAR4320101. RY and VS acknowledge NOAA grant NA16OAR4310100 and NSF grant 1748266. JL, GMW, RAH, JMS, and TFH acknowledge support from the NASA Tropospheric Composition Program and NOAA Climate Program Office's Atmospheric Chemistry, Carbon Cycle and Climate (AC4) program (NA17OAR4310004). DP, BAN, HG, PCJ, DAD, MKS, and JLJ were supported by NASA grants 80NSSC18K0630 and 80NSSC21K1451. AF was supported by NASA TCP Grant No. 80NSSC18K0628. The University of Innsbruck team was supported by the Austrian Federal Ministry for Transport, Innovation, and Technology (bmvit, FFG, ASAP). FP received funding from the European Union's Horizon 2020 research and innovation program under grant agreement No. 674911 (IMPACT EU ITN). LX, KTV, HA, JDC, and POW acknowledge NASA grant 80NSSC18K0660 and 80NSSC21K1704. This material is based upon work supported by the National Center for Atmospheric Research, sponsored by the National Science Foundation under Cooperative Agreement No. 1852977.

**Competing Interests:** AEP and SSB are contributing editors to Atmospheric Chemistry and Physics.

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
