# Peer review of "Parameterizations of US wildfire and prescribed fire emission"

_EGUsphere, 2023_

## Author Comment (AC1)

**Response to Reviewers**

**Parameterizations of US wildfire and prescribed fire emission ratios and emission factors based on FIREX-AQ aircraft measurements**
* * *
Referee comments are in **black** and authors responses are in **blue.**

**To editor and reviewers**:

We extend our gratitude for your valuable feedback and the time you dedicated to reviewing and enhancing the manuscript. In addition to your comments, in the revised version, we have corrected several data entries in Table S2 and S3. These corrections, while not affecting the primary conclusions of the paper, address the misnaming of certain fires. Additionally, we have revised one reference, one author's affiliation to reflect a change in their employment status since the original submission of this manuscript, and updated compound names in Table 3.

**Reviewer #1:**

This paper reports enhancement ratios (ERs) and "inferred" emission factors (EFs) for gas- and particle species measured in the smoke plumes of 9 western U.S. wildfires (and one eastern prescribed fire) during the 2019 Fire Influence on Regional to Global Environments and Air Quality (FIREX-AQ) campaign. The measurements were collected aboard the NASA DC-8 with what may be the most comprehensive instrument payload ever deployed for measuring atmospheric chemistry of biomass burning emissions. ERs and EFs are based on 16 cross-wind plume transects deemed to have the least photochemical aging based on measured ratios of furan, a fats-reacting primary biomass burning product, and maleic anhydride (a slow reacting secondary product). The paper focuses on emissions of non-methane organic gases (NMOG) and reactive nitrogen-containing compounds (NOy). In addition to ERs and EFs, the paper provides emission parameterizations for total NMOG and NOy which could be used with satellite observations of CO and NO2 to provide boundary conditions for atmospheric chemistry modeling.

Until very recently, detailed chemical speciation of relatively un-aged wildfire plumes in the western U.S. were unavailable. The 2013 SEAC4RS campaign provided the first such measurements, but for only a couple of wildfires. The wildfire focused 2018 Western Wildfire Experiment for Cloud Chemistry, Aerosol Absorption, and Nitrogen (WE-CAN) provided detailed chemical speciation of emissions and ERs and EFs for multiple western wildfires.

This paper builds upon these previous studies and further expands and improves our understanding of emissions from western U.S wildfires:

1. Additional measurements of highly variable process - The relative abundance of pollutants in "fresh" wildfire smoke is highly variable as it depends on fire behavior, fuels, and environmental conditions, all of which have high natural variability. Therefore, the additional ER and EF measurements provided in this study improve our characterization of emissions for western U.S. wildfires.
2. Expanded chemical speciation - The extensive payload allows for more detailed speciation of NMOG and nitrogen-containing species than provided by previous studies.

3. Reduced measurement uncertainty - NMOG were predominantly measured using four instruments: the NOAA proton transfer reaction time-of-flight mass spectrometer (PTR-ToF-MS), two whole air samplers (WAS) and the NOAA fast online gas chromatograph outfitted with a Time-of-Flight mass spectrometer (TOGA). Many of the NMOG species were measured by multiple instruments providing a means to reduce uncertainty.

4. Direct tie-in to extensive laboratory experiments - The NOAA PTR-ToF-MS was the same instrument used in FIREX 2016, the laboratory prequel to FIREX-AQ, provding a direct tie-in to the extensive lab measurements of biomass burning emissions form western U.S. fuels. This provides an improved interpretation of airborne measurements.

5. The study uses a "photochemical clock" to select least processed emissions - ERs and EFs from measurements of smoke that is less photochemically aged than previous studies, as demonstrated by the relative yields of primary and secondary products.

6. Parameterization of NMOG and NOy emissions - provides a potential pathway for employing satellite based measurements of total column CO and NO2 to estimate fire emissions for atmospheric modeling.

We thank the reviewer for the positive feedback and the helpful comments. Response to each comment is provided below.

**Minor Comments**

1. Table 1- Please described how the fuels were determined. Also, were the fuels extracted from area burned on day of flights?

   We now added at the caption: "Forest and shrubland fuel types were determined using the FCCS database, while cropland fires were classified with the Cropland Data Layer and DC-8 overflight videos (Warneke et al., 2022)."

2. The monoterpene ratio for FIREX-AQ/Firleab = 1.57. Did Firelab have better speciation of monoterpenes? If so, does this reflect differences in harvested vs. in-situ fuels?

   We attribute this difference primarily to emissions variability, as evidenced by the comparison of FIREX-AQ with previous campaigns in Table S6, which indicates higher Emission Factors (EFs) for this study in comparison to WE-CAN and Firelab by 2.3 and 1.6, respectively. There is agreement with SEAC4RS but lower EFs in comparison to Andreae 2019. Sekimoto et al. 2018 show that monoterpene emissions are largely driven by distillation processes, which occur early in the fire and differ from the high- and low-temperature pyrolysis reactions that impact the emissions of most other VOCs. We think that this difference could also explain the variability in monoterpene emissions in the field.

   Firelab did have higher speciation with some sampling by GC-PTR-ToF-MS; however, the ratios reported by Koss et al. 2018 and reported in Table S6 are based on the PTR-ToF-MS instrument also used in this study. Therefore, it is unlikely to be differences in instrument operation.

3. FIREX-AQ EF vs. WE-CAN EF (Table S6): FIREX-AQ/WE-CAN is 0.16 for HCN and 3.33 for HNCO. Can the authors provide any thought on possible reason for these large differences?

   These differences could occur for many reasons. First, HCN and HCNO are dependent on fuel nitrogen loadings (e.g., Coggon et al. 2016, Roberts et al. 2020), and so differences in fuel

composition could be a possible explanation. These emissions also have strong MCE relationships since HCN and HCNO are primary precursors for NOx in flaming fires (Roberts et al. 2020). Differences could also exist due to the challenges of measuring and calibrating for HCN and HNCO. HNCO and HCN exhibit strong, non-linear humidity dependencies within traditional PTR-ToF-MS instruments. For example, in the NOAA PTR-ToF-MS, HCN sensitivities at 80% RH are < 5% of the sensitivities measured under dry conditions. Consequently, at ambient conditions, the sensitivity is highly variable and likely very low for most plumes with moderate water content. HCN and HNCO measured during WE-CAN are also derived from a PTR-ToF-MS and likely faces similar challenges.

We note that this study, the emission factors reported for HCN are derived from the CF3O-CIMS, which was most sensitive to HCN. These measurements were also most reliable since the CF3O-CIMS carried an isotopically labeled HCN source in flight. Other measurements (including PTR-ToF-MS, I-CIMS, and TOGA) are in good agreement with the CF3O-CIMS (Table S5). HNCO was reported from the PTR-ToF-MS but has good agreement with the I-CIMS (Table S5).

4. Did the authors consider testing the ethyne/furan ratio based partitioning of NMOG emissions presented in Sekimoto et al. (2018) with the FIREX-AQ field data?

We felt that the partitioning of NMOG emissions was beyond the scope of this work. The Sekimoto et al. (2018) work was recently applied to the FIREX-AQ observations and shows that the laboratory partitioning of high- and low-temperature pyrolysis products is also applicable to field observations , but a third  factor for aging was appropriate. (https://pubs.acs.org/doi/full/10.1021/acs.est.3c00537).

5. L489-491: The dotted lines and shaded regions show FireLab parameterizations that describe how these ratios respond to changes in MCE (Roberts et al., 2020) for one fire burned during FireLab What fuel type in Firelab burn featured in Fig. 6? Are these curves indicative of typical Firelab burns?

This was a subalpine fir fire and is now incorporated into the main text. Similar systematic dependencies were indeed noted in Firelab burns, as detailed in the discussion by Roberts et al. (2020).

7. L255-256: Fire plumes sampled closest to the emission source that showed significant chemical processing with a MA/F > 0.20 are excluded from this analysis. How was the photochemical processing threshold of MA/F > 0.2 selected?

The ratio of 0.2 was selected based on the minimum slope of maleic anhydride to furan in all wildfire plumes given in Figure 2(b). Although, this could be considered arbitrary it's use here is solely to exclude processed plumes and use only the freshest crossing to calculate emission ratios and emissions factors during FIREX-AQ.

We have added additional text that points to Figure 2b as our justification for selecting this cut-off.

8. Given that maleic anhydride has multiple precursors in biomass smoke (other furans), that it is produced a couple reaction steps after the initial OH rxn, and that it has many production pathways, do the authors expect the relationship between MA/F ratio and photochemical activity be roughly the same across the 16 plumes? Could different emission rates/relative abundances of other furans and NOx or different photolysis environment significantly impact yields of MA?

We agree that the MA/F ratio may undergo changes downwind of a fire due to various factors such as NOx levels, furans distribution, and exposure to sunlight. In the context of the freshest plume

crossings where substantial MA formation has not yet occurred, we consistently observe similar slopes in the MA/F ratio (Figure 2b). Additionally, it's worth emphasizing that the distribution of furans is consistent across different Firelab experiments (Koss et al., 2018). This consistency suggests that despite variations in emission strength, the relative abundance of furans is likely less variable and thus, we suspect will result in similar production rates of MA. Further box modeling efforts are necessary to assess the sensitivity of this ratio downwind of wildfires comprehensively. However, for the objectives of this study, this ratio proves to be an effective and valuable metric for screening out significantly processed emissions.

We have added additional text to lines 233-234 pointing out that other furans contribute to MA, and that this proxy is only used here for screening out processed plumes.

9. L313: …uncertainties due to differences in calibration or canister effects are small. Table S5. In the case of furan, the WAS/NOAA PTR and iWAS/NOAA PTR agree with one another but are less than half the TOGA/NOAA PTR. This would seem to suggest that furan loss in the canisters is significant.

Certainly, canister uncertainties may contribute to the differences between WAS and iWAS versus TOGA. We have revised the text to omit "or canister effects," focusing instead on the possibility that the variability between PTR and GCs could originate from gases that cannot pass through a GC column. This includes the possibility of unidentified isomers and fragments from higher molecular weight species.

10. L593-595: We suggest that these differences could be due to differences in the fuel, quantification methods applied for each study, as well as due to differences in photochemical loss of reactive species prior to detection. Perhaps fire behavior and lofting of smoke (i.e. differential mix of flaming & smoldering) play a role as well.

This is a great point. We now added the following sentence: "Additionally, differences in fire behavior and the lofting of smoke, including variations in the mixture of flaming and smoldering combustion, could also be contributing factors."

**Technical Comments**

Figure 7 - Using fire names as labels makes it difficult to gauge location of actual measurement values on the plots. I suggest using markers instead.

We updated the figures to now include only markers.

Fig S4. Are the R2 for individual NMOG based on NMOG and CO medians of the 16 transects?

That is correct and is now included in the caption as follows: "The squares of correlation coefficients are calculated based on the linear regression fit applied to the medians derived from the 16 transects."

**Reviewer #2:**

This manuscript presents calculated emission ratios and emission factors from the FIREX-AQ campaign, comparing them to WECAN and the FIREX-AQ Laboratory emission factors. This work presents a detailed analysis of differences in the observed emissions between these campaigns due to instrumentation and smoke age. While not particularly exciting the manuscript is an important contribution to understanding wildfire emission factors and ratios based on the limitations of the instrumentation and field sampling design.

We thank the reviewer for the positive feedback and the helpful comments. Response to each comment is provided below.

**Specific Comments**

Line 154: "performed fast response in situ measurements" seems to include the whole air samplers which isn't accurate since those canisters are collected on the aircraft and analyzed later. Clarifying this point and adding addition information on how many of the WAS samples met the low photochemical processing threshold for the EF and ER calculations would be useful since the number of data points per flight is limited.

We removed "fast response in situ" from the text. The quantity of samples collected depends on the canister collection times that depends on the altitude. We now include in the main text the sampling time thresholds for each system.

Line 334: For the equation. What if there is no carbon in the compound (much of Figure 3b)?

The same equation applies but compounds with no carbon atoms do not contribute to the denominator of the equation $\sum_{x=1}^{n} \left( NC_x \cdot \frac{\Delta C_x}{\Delta CO} \right)$ given that $NC_x$ is zero.

Figure 7: It doesn't seem like the fire name is important to the figure (the fires aren't specifically referenced in the text) and just make the plots more difficult to read.

This is now updated to only include markers.

Table 3: In the text you discuss some compounds show better a relationship with NO2. Clarify in table 3 that the ERs are to CO. And consider adding a table (in text or to the supplement) of those species that should use NO2.

Added "ERs to CO". Species that correlate better with NO2 are currently shown in Figure S4.

**Technical Corrections:**

Figure 3: CO is too light and could be offset some to see the trace better. The overall quality of the figures should be improved.

We presume that the reviewer is referring to Figure 2. We have revised the figure to incorporate a darker color for CO.

Figure 5: it is very difficult to read these figure – the resolution seems poor and the size is rather small.

We further increased the size and quality of the figure.

Line 476: instead of "On the contrary" - In contrast

Done